# Ndc1 drives nuclear pore complex assembly independent of membrane biogenesis to promote nuclear formation and growth

Michael Sean Mauro[1], Gunta Celma[1], Vitaly Zimyanin[2,3], Magdalena M Magaj[2,3], Kimberley H Gibson[4], Stefanie Redemann[2,3,5], Shirin Bahmanyar[1]*

[1]Department of Molecular, Cellular and Developmental Biology, Yale University, New Haven, United States; [2]Center for Membrane and Cell Physiology, University of Virginia, Charlottesville, United States; [3]Department of Molecular Physiology and Biological Physics, University of Virginia, School of Medicine, Charlottesville, United States; [4]Center for Cellular and Molecular Imaging: Electron Microscopy, Department of Cell Biology, Yale School of Medicine, New Haven, United States; [5]Department of Cell Biology, University of Virginia, Charlottesville, United States

*For correspondence:
shirin.bahmanyar@yale.edu

Competing interest: The authors declare that no competing interests exist.

**Abstract** The nuclear envelope (NE) assembles and grows from bilayer lipids produced at the endoplasmic reticulum (ER). How ER membrane incorporation coordinates with assembly of nuclear pore complexes (NPCs) to generate a functional NE is not well understood. Here, we use the stereotypical first division of the early *C. elegans* embryo to test the role of the membrane-associated nucleoporin Ndc1 in coupling NPC assembly to NE formation and growth. 3D-EM tomography of reforming and expanded NEs establishes that Ndc1 determines NPC density. Loss of *ndc1* results in faster turnover of the outer scaffold nucleoporin Nup160 at the NE, providing an explanation for how Ndc1 controls NPC number. NE formation fails in the absence of both Ndc1 and the inner ring component Nup53, suggesting partially redundant roles in NPC assembly. Importantly, upregulation of membrane synthesis restored the slow rate of nuclear growth resulting from loss of *ndc1* but not from loss of *nup53*. Thus, membrane biogenesis can be decoupled from Ndc1-mediated NPC assembly to promote nuclear growth. Together, our data suggest that Ndc1 functions in parallel with Nup53 and membrane biogenesis to control NPC density and nuclear size.

## Editor's evaluation

The authors elegant studies help understand how the process by which membranes needed for growth of nuclear envelop is coordinated with nuclear pore assembly during nuclear envelop assembly. This coordination is mediated by a transmembrane nucleoporin Ndc1.

## Introduction

The nuclear envelope (NE) is a large double-membrane sheet that partitions the contents of the nucleus from the cytoplasm. The NE is continuous with the endoplasmic reticulum (ER), which produces the bilayer lipids from which the NE forms and expands (*Barger et al., 2022*). Nuclear pore complexes (NPCs), large multiprotein channels that control the bidirectional traffic of macromolecules across the NE, are embedded at fusion points between the inner and outer nuclear membranes (also known as the pore membrane; *Hetzer, 2010*). NPCs are composed of ~30 different proteins (nucleoporins

or Nups) that assemble in multiple copies to total ~1000 polypeptides (*Hampoelz et al., 2019a*; *Hampoelz and Baumbach, 2022*; *Lin and Hoelz, 2019*). Multiple NPC subcomplexes assemble into an eightfold symmetric core scaffold (the outer ring Nup107-160 complex and the inner ring Nup93 complex) (*Kosinski et al., 2016*; *Lin et al., 2016*) that surrounds the central channel and is attached to the cytoplasmic filaments and the nuclear basket, which are asymmetrically distributed (*Figure 1A*). A subset of integral membrane nucleoporins anchor NPCs to the pore membrane (*Hampoelz et al., 2019a*; *Lin and Hoelz, 2019*). Recent advances in EM approaches provide an unprecedented view of the architecture of this massive protein structure in human cells (*Schuller et al., 2021*; *Zila et al., 2021*) as well as its distinct assembly states at different cell cycle stages (*Otsuka and Ellenberg, 2018*; *Otsuka et al., 2018*). Intriguingly, membrane shape is not symmetric across the pore, raising questions about membrane sculpting activities by Nups for NPC function and assembly (*Lusk and King, 2021*).

In metazoans, the NE breaks down and reforms every cell division cycle (*Kutay et al., 2021*). NPCs disassemble into subcomplexes that then rapidly assemble onto segregated chromosomes as ER-derived membranes form the nuclear rim (*Hetzer, 2010*; *Kutay et al., 2021*). Nascent nuclear membranes initially cover the outer edges of chromatin where spindle microtubules (MTs) are less concentrated ('non-core'; *Liu and Pellman, 2020*; *Otsuka and Ellenberg, 2018*). The majority of NPCs assemble at the 'non-core' region of reforming NEs. NPC assembly is less pronounced in the 'core' region where the NE adaptor for the endosomal sorting complex required for transport (ESCRT), LEMD2/LEM-2, and the ESCRT II-III related protein CHMP-7 accumulates to facilitate severing of spindle MTs with sealing of remaining holes (*Gatta and Carlton, 2019*; *Kutay et al., 2021*; *Lusk and Ader, 2020*; *Zhen et al., 2021*). Once nuclear transport is established, nuclear import and recruitment of additional membranes promote nuclear growth (*Chen and Levy, 2022*). How NPC assembly is coupled to membrane recruitment is unclear; however, membrane-associated nucleoporins likely play a role (*De Magistris and Antonin, 2018*; *Kutay et al., 2021*; *Otsuka and Ellenberg, 2018*; *Ungricht and Kutay, 2017*).

There are three transmembrane-containing nucleoporins in metazoans: gp210 (*Gerace et al., 1982*), Ndc1 (*Mansfeld et al., 2006*; *Stavru et al., 2006*), and Pom121 (*Hallberg et al., 1993*) as well as several peripheral membrane binding Nups (*Hamed and Antonin, 2021*). gp210 is not ubiquitously expressed and is important for mammalian muscle differentiation (*D'Angelo et al., 2012*; *Raices et al., 2017*) and breakdown of the NE (*Galy et al., 2008*). In mammalian cells, Ndc1 and Pom121 are necessary for NPC biogenesis, possibly through redundant roles in membrane recruitment and NPC anchoring (*Mansfeld et al., 2006*). Pom121 has been also shown to be necessary for the interphase pathway of NPC assembly, which requires insertion of NPCs into an intact NE (*Doucet et al., 2010*).

After open mitosis, the chromatin-associated protein Elys initiates NPC assembly by recruitment of the outer ring scaffold (also known as Y-complex or Nup107/160 complex) onto chromatin (*Doucet et al., 2010*; *Franz et al., 2007*; *Rasala et al., 2006*). The recruitment and assembly of the Nup107/160 complex is an early, essential step in NPC assembly (*Walther et al., 2003*). In in vitro nuclear assembly reactions in *Xenopus* egg extracts, ER vesicles containing Ndc1 and Pom121 establish connections between reforming nuclear membranes and the Y-complex (*Rasala et al., 2008*). The recruitment of the inner ring component Nup53 (also referred to as Nup35) that is part of the Nup93 complex (*Lin and Hoelz, 2019*) initiates the assembly of the inner ring complex (*Eisenhardt et al., 2014*; *Vollmer et al., 2012*). Nup53 recruits Nup155, which assembles proximal to the pore membrane (*Dultz et al., 2008*; *De Magistris et al., 2018*; *Eisenhardt et al., 2014*; *Franz et al., 2005*; *Hampoelz et al., 2019a*; *Lin and Hoelz, 2019*; *Vollmer et al., 2012*). Nup155 and Nup53 interact with Pom121 and Ndc1 and also contain amphipathic helices that insert into membranes (*Hamed and Antonin, 2021*; *Mitchell et al., 2010*; *von Appen et al., 2015*). The interaction between Ndc1 and Nup53 is conserved in budding yeast (*Onischenko et al., 2009*); however, only the interphase pathway of NPC assembly exists in fungi because they undergo a closed mitosis (*Kutay et al., 2021*). Some evidence suggests a functional interaction between Ndc1 and Pom121 and members of the Nup107/160 complex (*Mitchell et al., 2010*), which also include Nups that directly bind to bilayer lipids (*Drin et al., 2007*; *Hamed and Antonin, 2021*). Thus, there are multiple connections that link the NPC to the pore membrane, yet their precise roles in NPC assembly remain poorly understood.

In addition to assembly of NPCs on chromatin through ELYS, recruitment of ER membranes with preassembled nucleoporins would effectively couple NPC assembly with nuclear formation and/or

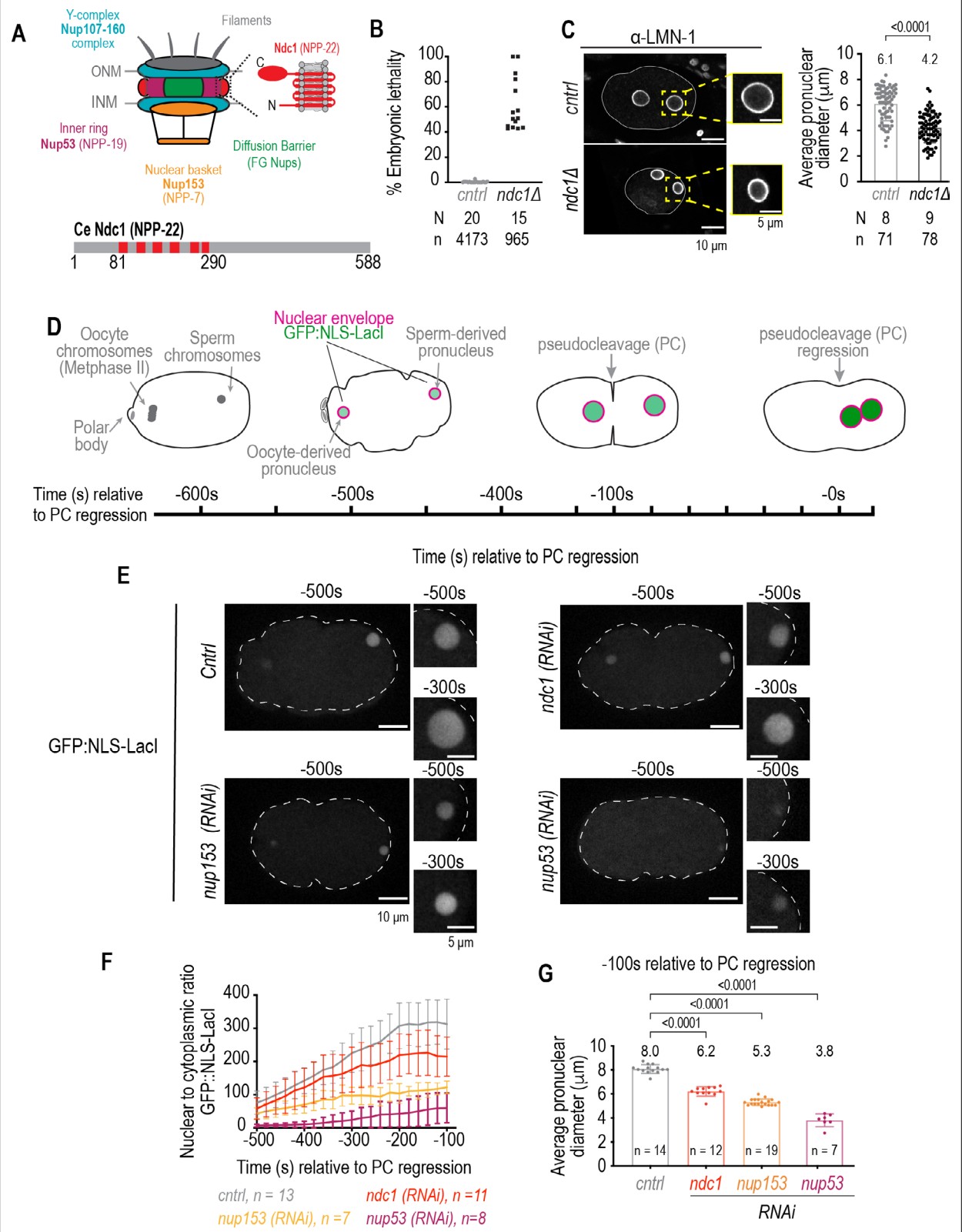

**Figure 1.** Smaller pronuclear size resulting from loss of Ndc1 corresponds to reduced nuclear import. (**A**) Schematic of a nuclear pore complex (left). Schematic and domain organization of *Ce* Ndc1 (right, bottom). (**B**) Plot of percentage embryonic lethality for indicated conditions. N = # of worms. n = # of embryos. (**C**) Left: fixed overview and magnified images of *C. elegans* embryos immunostained for lamin for indicated conditions. Scale bars, 10 µm and 5 µm for magnified images. Right: plot of nuclear size for indicated conditions. Mean ± S.D. N = # of slides, n = # of nuclei. (**D**) Schematic of

*Figure 1 continued on next page*

*Figure 1 continued*

stereotypical nuclear events relative to nuclear import in the *C. elegans* zygote, with pseudocleavage (PC) regression used as a reference time point. Time is in seconds. (**E**) Confocal overview and magnified images of embryo from a time lapse series of GFP:NLS-LacI in indicated conditions. Scale bars, 10 μm for overview image and 5 μm for magnified images. (**F**) Plot of nuclear to cytoplasmic ratio of GFP:NLS-LacI for indicated conditions. n = # of embryos. Average ± S.D. is shown. (**G**) Pronuclear diameter for indicated conditions at indicated time point. Average ± S.D. is shown. n = # of embryos. A two-way ANOVA was used to determine statistical significance between control and each RNAi condition. p-Values all <0.0001.

The online version of this article includes the following video, source data, and figure supplement(s) for figure 1:

**Source data 1.** Excel file containing individual data points related to *Figure 1*.

**Figure supplement 1.** Generation and characterization of mutant *ndc1* alleles and RNAi depletion of *ndc1*.

**Figure supplement 1—source data 1.** Excel file containing individual data points.

**Figure 1—video 1.** Nuclear import scales with nuclear size in one-cell stage *C. elegans* embryos.

https://elifesciences.org/articles/75513/figures#fig1video1

expansion (*Hampoelz and Baumbach, 2022*; *Kutay et al., 2021*). Recent evidence in mammalian cells showed that mitotic ER membranes contain partially preassembled nuclear pores that initiate NPC assembly after chromosome segregation (*Chou et al., 2021*). In support of this, ultrastructural work showed that the NE forms from highly fenestrated ER membrane sheets (*Otsuka et al., 2018*) that could contain some nucleoporins. In *Drosophila* oocytes, nucleoporin condensates in the cytoplasm form the nuclear pore scaffold in a specialized ER domain called annulate lamellae (*Hampoelz et al., 2019b*). These annulate lamellae feed NE expansion facilitating NPC insertion with rapid nuclear growth (*Hampoelz et al., 2016*). The molecular link between preassembled nucleoporins and ER membranes is not known but may involve transmembrane-containing Nups.

Here, we use the stereotypical first division of *C. elegans* embryos to understand the role of Ndc1 in coupling NPC biogenesis with ER membrane recruitment during nuclear formation and expansion. The stereotypical nuclear events in the first embryonic division make the early *C. elegans* embryo an ideal system to study the direct effects of Ndc1 depletion on the first attempt at nuclear formation and expansion. Furthermore, Pom121 is not present in *C. elegans* providing the opportunity to understand the specific contribution of Ndc1 in NPC biogenesis independent of Pom121. Prior work has shown that Ndc1 is only partially essential in *C. elegans* and impacts NPCs, yet the focus on late-stage embryos made it difficult to parse out direct versus accumulated effects due to multiple rounds of division (*Stavru et al., 2006*).

We show that loss of Ndc1 results in small nuclei with reduced NPCs and nuclear import rates in early *C. elegans* embryos. We use a mutant strain in *cnep-1*, a negative regulator of ER and nuclear membrane biogenesis (*Bahmanyar et al., 2014*; *Bahmanyar and Schlieker, 2020*), to show that increased membrane biogenesis results in faster NE expansion. Increased membrane biogenesis partially suppressed the slow rate of nuclear expansion, but not reduced nuclear import and NPC density, that results from loss of *ndc1*. Importantly, upregulated membrane biogenesis did not restore the small size of nuclei resulting from loss of *nup53* or *nup153* suggesting that membrane production requires these nucleoporins but does not require Ndc1 to drive nuclear expansion. We show that Ndc1 controls NPC density by promoting stable incorporation of outer ring scaffold components. Genetic analyses reveal that *nup53* and *ndc1* play parallel roles in NPC assembly, possibly through Nup155, an essential linker of the NPC scaffold (*Lin and Hoelz, 2019*; *von Appen et al., 2015*). Together, our data suggest that Ndc1-mediated NPC assembly functions in parallel to Nup53 and membrane biogenesis to control NPC density and nuclear size.

## Results

### Nuclear size correlates with nuclear import rates in early *C. elegans* embryos

A genome-wide RNAi screen in *C. elegans* embryos revealed that RNAi depletion of the transmembrane nucleoporin NPP-22 (hereafter Ndc1; *Figure 1A*; for a detailed list of *C. elegans* nucleoporins and their human homologs see *Galy et al., 2003*) results in small pronuclei prompting us to test if deletion of the entire *ndc1* gene locus by CRISPR-Cas9 gene editing to eliminate *ndc1* mRNA expression would cause a more severe nuclear phenotype (*Figure 1—figure supplement 1A* and

*Figure 1—figure supplement 1B*; *ndc1Δ*; *Sönnichsen et al., 2005*). Embryos produced from homozygous *ndc1Δ* worms had a range of lethality (40–100%, average ± S.D.=61 ± 21%, *Figure 1B*) that was similar but more severe than embryonic lethality caused by a mutation in *ndc1* (*ndc1^tm1845^*) in which 50% of the *ndc1* gene coding region is deleted (range 10–80%, average ± S.D.=48 ± 25%, *Figure 1—figure supplement 1D* and *Figure 1—figure supplement 1E*; *Stavru et al., 2006*). The length of a subset of *ndc1Δ* embryos, but not *ndc1* RNAi-depleted embryos, was on average smaller (44% 0.75 times average control embryo length, N=9 slides n=78 embryos) suggesting a potential defect in germline development that may explain the embryonic lethality that was not observed by RNAi depletion of *ndc1* (*Figure 1C*, *Figure 1—figure supplement 1C*, and *Figure 1—figure supplement 1F*). *ndc1Δ* embryos contained small pronuclei (*Figure 1C*) that were similar in size to those RNAi-depleted for *ndc1* (*Figure 1—figure supplement 1C*) suggesting that Ndc1 determines nuclear size in early embryos.

Nuclear size is correlated with nuclear import rates (*Levy and Heald, 2010*; *Levy and Heald, 2012*). We quantitatively monitored nuclear import in *C. elegans* embryos from the onset of nuclear formation after fertilization until mitotic entry (*Figure 1D*; *Penfield et al., 2020*). Fertilization of the oocyte by haploid sperm initiates two rounds of meiotic chromosome segregation. The sperm chromatin is devoid of a NE at the time of fertilization. After anaphase of meiosis II, components in the oocyte cytoplasm are recruited to the sperm-derived pronucleus to form a functional NE (*Figure 1D*), a process analogous to NE reformation after mitosis. Transport competent pronuclei then expand ~30-fold in volume prior to the onset of nuclear permeabilization in mitosis, which occurs after pronuclear meeting and regression of the pseudocleavage (hereafter known as PC regression), thus providing an assay for measuring nuclear import rates directly following the first attempt at NE formation (*Figure 1D*; *Oegema and Hyman, 2006*).

Nuclear import monitored by a GFP reporter fused to a nuclear localization signal (GFP:NLS-LacI, hereafter referred to as GFP:NLS) shows nuclear accumulation of GFP fluorescence throughout the time course of nuclear expansion (*Figure 1D-F* and *Figure 1—figure supplement 1G*). RNAi depletion of Ndc1 results in lower levels of nuclear accumulation of the GFP:NLS reporter (*Figure 1E-F*, *Figure 1—video 1*). Nuclear import is more severely impaired in embryos depleted of the nuclear basket component Nup153 (*npp-7*, *Figure 1A*) and the essential inner ring component Nup53 (*npp-19*; *Figure 1A, E and F*, and *Figure 1—video 1*; *Galy et al., 2003*; *Lin and Hoelz, 2019*). The average diameter of pronuclei is ~8 µm ~ −100 s relative to PC regression in control embryos (*Figure 1G*), and the diameter of pronuclei at the same time point in each RNAi condition scales with the degree of severity in defective nuclear import for that condition (6.2 µm for *ndc1* [*RNAi*] versus 5.3 µm for *nup153* [*RNAi*] and 3.8 µm for *nup53* [*RNAi*]; *Figure 1F and G*). Thus, loss of distinct components of NPCs results in nuclear size defects that scale with nuclear import rates.

## Delayed onset of nuclear import results from loss of *ndc1*

Slower nuclear import rates may arise from defects in NPC assembly during nuclear formation and/or expansion. We found that Ndc1 fused to mNeonGreen at its C-terminus using CRISPR-Cas9 gene editing (Ndc1^en^:mNG, *Figure 2—figure supplement 1A*; *Figure 2—figure supplement 1B*; *Figure 2—video 1*) appears to accumulate at non-core regions of the reforming NE after mitosis with similar kinetics as the inner NE protein LEM-2, which enriches at the core regions (*Figure 2A and B*, and *Figure 2—video 2*), as has been described in mammalian cells (*Liu and Pellman, 2020*), and coincident with the appearance of signal from the fluorescent ER maker SP12:mCh around segregated chromosomes (*Figure 2—figure supplement 1D*). Ndc1:mNG also localizes to punctate structures throughout the cytoplasm that mostly disperse into the ER upon entry into mitosis (*Figure 2—figure supplement 1C*). The early recruitment of Ndc1 to the reforming NE suggested a potential role for Ndc1 in post-mitotic nuclear assembly to establish nuclear import.

We analyzed the kinetics of NE formation relative to the establishment of nuclear import directly after anaphase onset in the nucleus of the larger daughter cell (also cell known as the 'AB' cell based on the *C. elegans* lineage map). The NE rapidly forms around daughter nuclei ~100 s after anaphase onset, coincident with the ingression of the cytokinetic furrow and followed by expansion of daughter nuclei (*Figure 2A and C*). In control embryos, the GFP:NLS fluorescence signal appears in the nucleus ~40 s following initiation of furrow ingression, which is ~60–80 s following the initial presence of a complete nuclear rim marked by LEM-2:mCh (–20 s, *Figure 2C*). GFP:NLS continues to accumulate

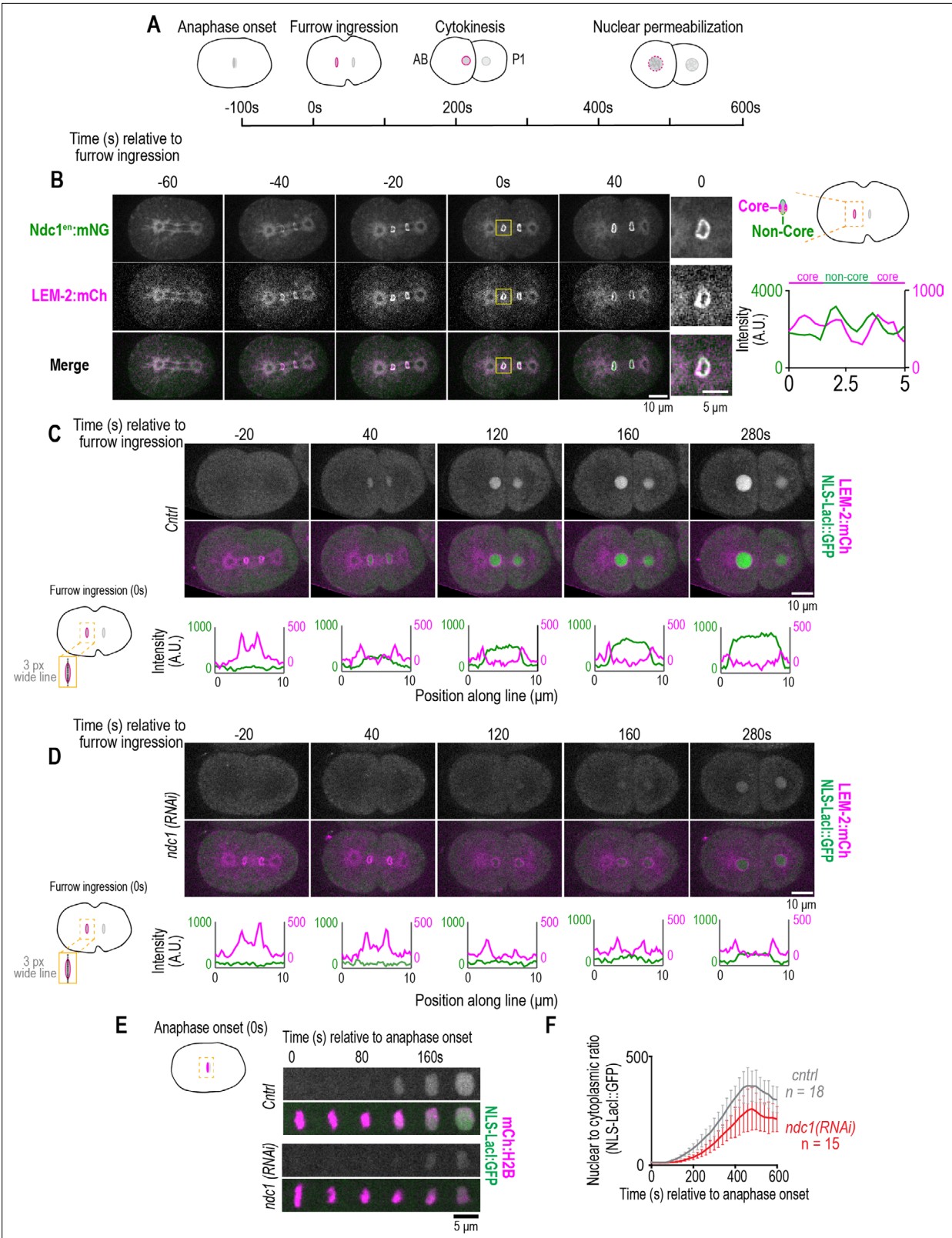

**Figure 2.** Ndc1 is necessary for timely formation of a transport competent nucleus after mitosis. (**A**) Schematic of the first mitotic division in *C. elegans* embryos relative to anaphase onset and initiation of furrow ingression. All measurements of one- to two-cell stage embryos are done on the AB cell/nucleus (nuclear envelope [NE] of the AB nucleus is highlighted in magenta). (**B**) (Left) Confocal images from time series of mitotic nuclear formation relative to furrow ingression with indicated markers. (Right) Three-pixel wide line scan for the 'core' and 'non-core' region of the NE. Scale bars, 10 µm

*Figure 2 continued on next page*

*Figure 2 continued*

for overview image and 5 µm for magnified images. (**C–D**) (Above) Confocal images from time series mitotic nuclear formation with indicated markers for indicated conditions. (Below) Line scans measuring background-corrected fluorescence intensities as indicated in schematic for each time point and fluorescent marker (LEM-2:mCH is magenta and NLS-LacI:GFP is green). Scale bars, 10 µm and 5 µm for magnified images. (**E**) Confocal images of chromosome region from time series relative to anaphase onset with indicated markers and in indicated conditions. Scale bar, 5 µm. (**F**) Plot of nuclear to cytoplasmic ratio of GFP:NLS-LacI for indicated conditions. n = # of embryos. Average ± S.D. is shown.

The online version of this article includes the following video, source data, and figure supplement(s) for figure 2:

**Source data 1.** Excel file containing individual data points related to *Figure 2.*

**Figure supplement 1.** Ndc1^en:mNG is recruited early to the nuclear rim and localizes to the endoplasmic reticulum (ER) and cytoplasmic puncta, and nuclear import is delayed in post-mitotic nuclei without Ndc1.

**Figure supplement 1—source data 1.** Excel file containing individual data points.

**Figure 2—video 1.** Endogenously tagged Ndc1:mNG is enriched at the nuclear rim and produces embryos that progress normally through the one-cell stage.

https://elifesciences.org/articles/75513/figures#fig2video1

**Figure 2—video 2.** Recruitment of endogenously tagged Ndc1:mNG relative to the inner nuclear envelope (NE) marker LEM-2:mCherry.

https://elifesciences.org/articles/75513/figures#fig2video2

**Figure 2—video 3.** Loss of Ndc1 delays the initiation of nuclear transport after mitosis.

https://elifesciences.org/articles/75513/figures#fig2video3

in the nucleus as nuclei expand (*Figure 2C*). In the absence of *ndc1*, LEM-2:mCh forms a nuclear rim with normal timing (approximately 20–40 s prior to cleavage furrow ingression); however, the GFP:NLS fluorescence signal is detectable appreciably later in daughter nuclei relative to cleavage furrow ingression compared to control (control: average ± S.D.=3.1 s±10.7 s, n=13 embryos; *ndc1* RNAi-depleted embryos: average ± S.D.=82 s±47 s, n=15 embryos; *Figure 2D*). The nuclear to cytoplasmic ratio of GFP:NLS fluorescence signal in nuclei marked by mCh:Histone2B was quantified after anaphase onset and indicated a delay and reduction in GFP:NLS accumulation in *ndc1* RNAi-depleted embryos, which was followed by a decrease in signal at mitotic-entry-induced nuclear permeabilization in both control and *ndc1* RNAi-depleted embryos (*Figure 2E and F*; *Figure 2—video 3*). Normalization of the traces and plotting the difference of the average normalized values between each time point further revealed a shift in the onset of nuclear accumulation of GFP:NLS in *ndc1* RNAi-depleted embryos (*Figure 2—figure supplement 1E* and *Figure 2—figure supplement 1F*). Together, these data show that Ndc1 is necessary for the timely establishment of nuclear import, but not for the bulk recruitment of membranes, during nuclear reformation.

## Ndc1 mutants contain fewer nuclear pores upon nuclear reformation and in expanded nuclei

We predicted that Ndc1 is necessary for timely nuclear accumulation of the GFP:NLS reporter by promoting NPC assembly. We analyzed serial sections from electron tomograms of NE formation in a control and *ndc1Δ* embryo processed at the initiation of furrow ingression (*Figure 3A and B*; *Figure 3—videos 1–3*). We focused on nascent nuclear membranes wrapped around the outer edges of chromatin or 'non-core' region (*Liu and Pellman, 2020*). 3D analysis of the reforming NE revealed small and large gaps at this time point and those <100 nm were marked as potential NPCs (*Figure 3A and B*; *Figure 3—videos 4; 5*; *Otsuka et al., 2018*). The NE in the *ndc1Δ* embryo was mostly continuous and contained an average of 11 'NPC' holes per µm² (N=1 nuclei, n=4 areas) (*Figure 3B*, *Figure 3—figure supplement 1B* and *Figure 3—video 5*). In contrast, the NE in a control embryo contained an average of 55 'NPC' holes per µm² (N=1 nuclei, n=4 areas) and was more discontinuous (*Figure 3A and C*; *Figure 3—video 4*). Thus, when *ndc1* is absent, nascent NEs of the 'non-core' region are more continuous and on average contain ~4.8-fold fewer holes that fit the dimensions of nascent NPCs (*Otsuka et al., 2018*).

Fully formed NEs in two- to four-cell stage embryos contained half the normal density of NPCs in *ndc1Δ* mutants. Control nuclei on average contained 51 'NPC' holes per µm² (N=2 nuclei, n=6 areas), while *ndc1Δ* nuclei contained 26 'NPC' holes per µm² (N=3 nuclei, n=9 areas) (*Figure 3—figure supplement 1A* and *Figure 3—figure supplement 1B*, *Figure 3—video 6*). Thus, in both fully formed and reforming NEs in *ndc1Δ* embryos the density of NPCs is reduced.

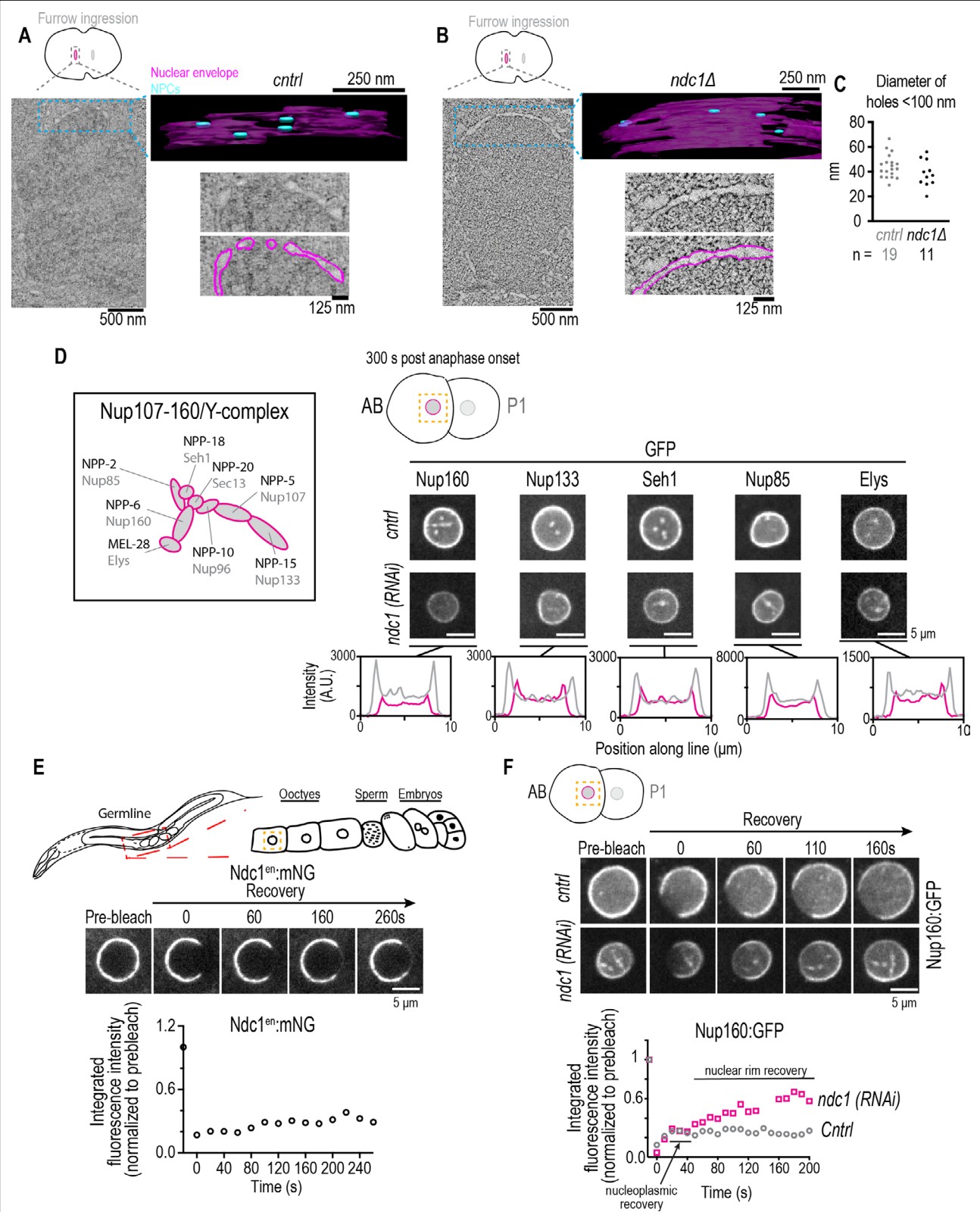

**Figure 3.** Fewer nuclear pore complexes (NPCs) assembled on nascent nuclear envelopes (NEs) and a higher mobile pool of Nup160:GFP result from loss of *ndc1*. (**A, B**) Overview images: z-slice from electron tomogram of nuclear formation timed relative to initiation of furrow ingression. 3D model: traced membranes (magenta) and NE holes <100 nm (blue) for region shown as single z-slice in overview image. Magnified representative z-slice traced and untraced from electron tomogram is shown. To calculate the density of 'NPC' holes during NE reformation, we segmented and quantified four areas

*Figure 3 continued on next page*

*Figure 3 continued*

from a single control embryo (0.14 µm², 0.08 µm², 0.05 µm², and 0.04 µm²) and four areas from a single *ndc1Δ* embryo (0.22 µm², 0.19 µm², 0.18 µm², and 0.17 µm²). Scale bars indicated in figure. (**C**) Plot of diameters of NE holes <100 nm. n = # of 'NPC' holes analyzed. (**D**) Schematic of Y-complex (also known as Nup107/160 complex) with vertebrate (gray) and *C. elegans* (black) names shown, adapted from Figure 1B in *Hattersley et al., 2016* (note that NPP-23/Nup43 not included). Representative magnified images of AB nucleus from confocal series for each indicated marker (above) and line scans measuring background-corrected fluorescence intensities (below) for each condition. Scale bars, 5 µm. (**E**) (Top) Schematic of *C. elegans* gonad and oocytes. (Middle) Confocal images from time lapse series of fluorescence recovery after photobleaching (FRAP) of Ndc1^en^:mNG the NE of an oocyte is shown. (Bottom) Representative plot of fluorescence intensities of bleached region over time normalized to the prebleach intensity for each condition is shown. Scale bars, 5 µm. (**F**) Confocal images from time lapse series of FRAP of Nup160:GFP at the NE of an embryo for indicated conditions are shown (above). Representative plot of fluorescence intensity of bleached region over time normalized to the prebleach intensity for each condition is shown (below). Scale bars, 5 µm.

The online version of this article includes the following video, source data, and figure supplement(s) for figure 3:

**Source data 1.** Excel file containing individual data points related to *Figure 3.*

**Figure supplement 1.** Reduced nuclear pore complex density in expanded nuclei, levels of outer ring scaffold nucleoporins at the nuclear rim and cytoplasmic puncta resulti from loss of Ndc1.

**Figure supplement 1—source data 1.** Excel file containing individual data points.

**Figure supplement 1—source data 2.** Source image for the immunoblot in *Figure 3—figure supplement 1G* (Nup107 levels).

**Figure supplement 2.** Characterization of protein levels and localization of nucleoporins in *ndc1* deletion mutant and fluorescence recovery after photobleaching (FRAP) analysis of Ndc1^en^:mNG in nuclear envelope (NE) of embryos.

**Figure supplement 2—source data 1.** Excel file containing individual data points.

**Figure supplement 2—source data 2.** Source image for the immunoblots (*Figure 3—figure supplement 2A-D*).

**Figure 3—video 1.** Electron tomogram of control nucleus from embryo frozen at onset of furrow ingression, z sections 315–386 (165 nm total). https://elifesciences.org/articles/75513/figures#fig3video1

**Figure 3—video 2.** Electron tomogram of control nucleus from embryo frozen at onset of furrow ingression z sections 524–595 (165 nm total). https://elifesciences.org/articles/75513/figures#fig3video2

**Figure 3—video 3.** Electron tomogram of ndc1Δ nucleus from embryo frozen at onset of furrow ingression z sections 8–309 (495 nm total). https://elifesciences.org/articles/75513/figures#fig3video3

**Figure 3—video 4.** 3D model of region of tomogram in *Figure 3—videos 1; 2*. https://elifesciences.org/articles/75513/figures#fig3video4

**Figure 3—video 5.** 3D model of region of tomogram in *Figure 3—video 3*. https://elifesciences.org/articles/75513/figures#fig3video5

**Figure 3—video 6.** Electron tomogram of control and ndc1Δ nucleus from interphase embryo; z sections 150–250 (212 nm total) for control and z sections 30–130 (212 nm total) for ndc1Δ, related to Figure 3—figure supplement 1. https://elifesciences.org/articles/75513/figures#fig3video6

**Figure 3—video 7.** Ndc1 is required for stable incorporation of Nup160:GFP. https://elifesciences.org/articles/75513/figures#fig3video7

**Figure 3—video 8.** Fluorescence recovery after photobleaching of Ndc1:mNG at the nuclear envelope (NE) of expanding one- to two-cell stage nucleus. https://elifesciences.org/articles/75513/figures#fig3video8

We reasoned that if *ndc1* RNAi depletion results in fewer nuclear pores in the NE, this should be reflected in the fluorescence intensity of members of the Nup107-160 complex, also known as the Y-complex (*Figure 3D*; *Hampoelz et al., 2019a*; *Lin and Hoelz, 2019*), as well as other components of mature NPCs. In early *C. elegans* embryos, the Y-complex component Nup160:GFP localizes to kinetochores and to the nuclear rim (*Hattersley et al., 2016*). Nup160:GFP also localizes to puncta throughout the cytoplasm that occasionally co-localize with ER-associated Ndc1 puncta and disperse upon entry into mitosis (*Figure 3—figure supplement 1C* and *Figure 3—figure supplement 1D*). *ndc1* RNAi-depleted embryos contained lower levels of Nup160:GFP at the nuclear rim (*Figure 3D*) and very few puncta formed in the cytoplasm (*Figure 3—figure supplement 1D*). The fact that they co-localize with Ndc1 and do not form in *ndc1* RNAi embryos, which also have lower levels of Nup160:GFP at the nuclear rim, suggested that Ndc1 may play a role in stabilizing the outer scaffold components both in these cytoplasmic structures and in the NE.

The reduced nuclear rim signal of Nup160:GFP resulting from loss of *ndc1* likely reflects the lower density of NPCs in the NE. In addition to Nup160:GFP, line profiles of fluorescence intensities of GFP fusions to other members of the Y-complex (Nup133, Seh1, Nup85, and Elys; *Hattersley et al., 2016*) revealed that the peak fluorescence signal of each in *ndc1* RNAi-depleted nuclei is approximately half that of control (*Figure 3D*). Immunostaining with antibodies that recognize Nup107 and Elys confirmed lower endogenous levels of Y-complex nucleoporins at the nuclear rim in *ndc1Δ* early embryos (*Figure 3—figure supplement 1E* and *Figure 3—figure supplement 1F*; *Galy et al., 2006*; *Ródenas et al., 2012*). The total protein levels of Nup107 in *ndc1Δ* worms were similar to control worms (*Figure 3—figure supplement 1G*). Deletion of *ndc1* also results in lower levels of the endogenous inner ring component Nup53 at the nuclear rim (*Figure 3—figure supplement 2A*) but not its global protein levels (*Figure 3—figure supplement 2B*). Immunostaining of control and *ndc1* mutant worms with mAB414, a monoclonal antibody that recognizes FG-nucleoporins, also showed a reduction in fluorescence signal at the nuclear rim (*Figure 3—figure supplement 2C*), as has been shown for late-stage *C. elegans* embryos (*Stavru et al., 2006*) and in vertebrate cells (*Mansfeld et al., 2006*). Total protein levels of mAB414 epitope-containing nucleoporins as well as Nup96 and Importin α3, which are expressed in both the germline and somatic cells in *C. elegans* (*Geles and Adam, 2001*), were unchanged (*Figure 3—figure supplement 2D*, *Figure 3—figure supplement 2E* and *Figure 3—figure supplement 2F*). Together, these results provide evidence confirming our EM tomography data that loss of Ndc1 reduces NPC density.

## Ndc1 is necessary for immobilization of the outer ring scaffold in the NE

It has been proposed that transmembrane nucleoporins serve as an anchor to immobilize the NPC in the NE. In nuclei that were expanding, fluorescence recovery after photobleaching (FRAP) revealed that Ndc1$^{en}$:mNG is highly mobile in the NE (*Figure 3—figure supplement 2G*; average mobile fraction ± S.D.=0.59 ±- 0.09, n=7 embryos). This suggests that Ndc1$^{en}$:mNG may dynamically associate with nascent NPCs. However, growth of nuclei could result in recovery of Ndc1$^{en}$:mNG through feeding of a new pool of membrane-associated Ndc1 and so we tested the turnover of Ndc1$^{en}$:mNG in the fully expanded nuclei of oocytes. Ndc1$^{en}$:mNG turnover was slow in oocyte NEs indicating that Ndc1$^{en}$:mNG was immobile in mature NPCs (*Figure 3E*; average mobile fraction ± S.D.=0.22 ±- 0.2, n=9 embryos). Thus, NDC1 may serve an anchoring role for mature NPCs.

We reasoned that if Ndc1 is an anchor to nuclear pores, then the lower signal of Nup107/160 in *ndc1* RNAi-depleted embryos may result from increased turnover of the Nup107/160 complex in the NE. FRAP revealed that ~80% of the Nup160:GFP pool in the NE is immobile in control embryos (*Figure 3F* and *Figure 3—video 7*; average mobile fraction ± S.D.=0.20±0.06, n=8 nuclei), similar to what has been shown in mammalian cells (*Rabut et al., 2004*). In *ndc1* RNAi-depleted embryos, there was a greater than twofold increase in the mobile fraction of Nup160:GFP in the NE indicating that Nup160:GFP is less stably incorporated without Ndc1 (*Figure 3F* and *Figure 3—video 7*; *Figure 3—video 8* average mobile fraction ± S.D.=0.47±0.12, n=8 nuclei). We conclude that Ndc1 is necessary to immobilize the outer ring scaffold to promote stable NPC assembly during nuclear formation and growth.

## Increasing lipid synthesis in *ndc1* mutants restores nuclear growth but not reduced Nup160 levels in the NE or nuclear import rates

We next tested if increasing ER/NE membrane biogenesis was sufficient to restore nuclear formation and nuclear growth rates of *ndc1* mutants despite the decrease in nuclear pore biogenesis and nuclear import. We focused on the growth phase of nuclei in one and one- to two-cell stage embryos because of the stereotypic and reproducible rates of nuclear expansion. We first confirmed the spherical shape of pronuclei in *C. elegans* embryos as they expand by isotropic fluorescence light sheet imaging (*Figure 4A* and *Figure 4—video 1*). We then used semi-automated diameter measurements of the central section of nuclei during the expansion phase to extrapolate the rate of expansion of nuclear volume in control and *ndc1* mutant embryos (*Figure 4A-D*, *Figure 4—figure supplement 1A-C*). A significantly slower rate of expansion of nuclear volume reflected the smaller nuclear size of *ndc1* mutant embryos (see also *Figure 1*).

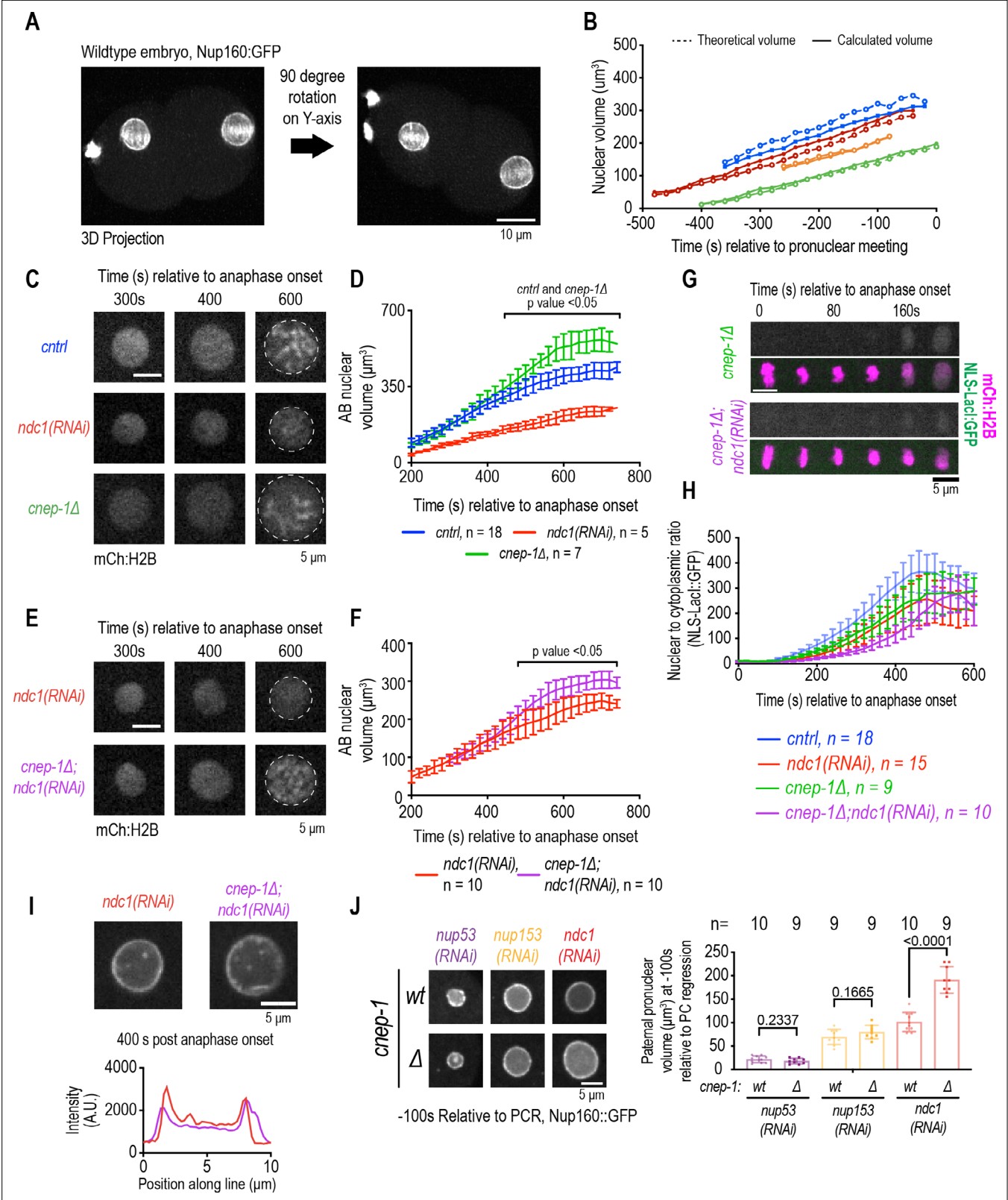

**Figure 4.** Increasing membrane biogenesis restores the slow rate of nuclear expansion and small nuclear size resulting from loss of *ndc1*. (**A**) Dual-view inverted light sheet microscopy 3D projection of wild-type one-cell stage embryo. Scale bar, 10 μm. (**B**) Theoretical and calculated pronuclear volume for indicated time points. Each color represents a distinct embryo. (**C**) Magnified images of AB nucleus from confocal series for mCh:H2B for indicated conditions. Scale bar, 5 μm. (**D**) AB nuclear volume for indicated conditions at indicated time points. Average ± S.D. is shown. (**E**) Magnified images of

*Figure 4 continued on next page*

*Figure 4 continued*

AB nucleus from confocal series for mCh:H2B for indicated conditions. Scale bar, 5 μm. (**F**) AB nuclear volume for indicated conditions at indicated time points. Average ± S.D. is shown. (**G**) Confocal images of chromosome region from time series relative to anaphase onset with indicated markers and in indicated conditions. Scale bar, 5 μm. (**H**) Plot of nuclear to cytoplasmic ratio of GFP:NLS-LacI for indicated conditions. n = # of embryos. Average ± S.D. is shown. (**I**) Magnified images of AB nucleus from confocal series for Nup160:GFP (above) and line scans measuring background-corrected fluorescence intensities (below) for each condition. Scale bar, 5 μm. (**J**) (Left) Magnified images of paternal pronucleus at –100 s relative to pseudocleavage (PC) regression expressing Nup160:GFP under indicated conditions. Scale bar, 5 μm. (Right) Plot of average pronuclear volume for indicated conditions at –100 s relative to PC regression. n = # of embryos.

The online version of this article includes the following video, source data, and figure supplement(s) for figure 4:

**Source data 1.** Excel file containing individual data points related to *Figure 4.*

**Figure supplement 1.** Characterization of nuclear size in embryos lacking *ndc1* and *cnep-1*.

**Figure supplement 1—source data 1.** Excel file containing individual data points.

**Figure 4—video 1.** Dual-view inverted light sheet microscopy 3D reconstructions of wild-type and cnep-1Δ embryos, related to Figure 4 and Figure 4—figure supplement 1.

https://elifesciences.org/articles/75513/figures#fig4video1

Deletion of *cnep-1,* a negative regulator of ER membrane biogenesis in *C. elegans* (*Bahmanyar et al., 2014*; *Penfield et al., 2020*), resulted in a faster rate of isotropic nuclear growth even with lower levels of the NLS reporter retained in the nucleus, which we had previously showed was due defective NE closure (*Penfield et al., 2020*; *Figure 4C, D and H* and *Figure 4—figure supplement 1B*, *Figure 4—figure supplement 1C*). Increasing membrane biogenesis in *ndc1* RNAi-depleted embryos deleted of *cnep-1* partially restored the rate of nuclear growth to closer to wild-type levels (*Figure 4E and F*, *Figure 4—figure supplement 1D*, and *Figure 4—figure supplement 1E*). Deletion of *cnep-1* did not restore the decreased fluorescence signal of Nup160:GFP in the NE (*Figure 4I*) nor the absence of cytoplasmic Nup160:GFP puncta (*Figure 4—figure supplement 1F* and *Figure 4—figure supplement 1G*) resulting from loss of *ndc1*. Interestingly, *cnep-1* deletion did not restore the small nuclear size resulting from loss of *nup53* or *nup153* (*Figure 4J*). Thus, membrane-mediated nuclear expansion can be decoupled from Ndc1-dependent nuclear pore biogenesis.

Regulation of lipid synthesis along with NE-specific ESCRT-membrane remodeling machinery promotes closure of NE holes and restricts incoming membranes to the chromatin surface (*Penfield et al., 2020*). Thus, the additive delay in nuclear formation may reflect the independent requirements for CNEP-1 and Ndc1 to establish timely transport competence through closure of NE holes and NPC biogenesis, respectively. In support of this, deletion of the ESCRT-III adaptor *chmp-7* in *ndc1* RNAi-depleted embryos also resulted in a delay in nuclear import (*Figure 4—figure supplement 1H*).

## Ndc1 supports nuclear formation and growth through a pathway that is in parallel to Nup53

Ndc1 binds directly to Nup53, and both bind to Nup155 (*Eisenhardt et al., 2014*; *Mitchell et al., 2010*; *Vollmer et al., 2012*) and have a role in assembly of the inner ring scaffold (*Mansfeld et al., 2006*; *Vollmer et al., 2012*; *Figure 5—figure supplement 1A*). We utilized a *C. elegans* strain carrying a partially functional mutant allele of Nup53 (*nup53*[tm2886]) that is missing a central region (Δaa 217–286), which would be predicted to disrupt its dimerization, membrane binding, and association to Nup155 (*Figure 5—figure supplement 1B*; *De Magistris et al., 2018*; *Eisenhardt et al., 2014*; *Ródenas et al., 2009*; *Vollmer et al., 2012*), to determine if Ndc1 and Nup53 are in the same pathway to promote post-mitotic NPC assembly in vivo. The Nup53[tm2886] mutant protein is expressed at lower levels compared to wild-type Nup53 (*Figure 3—figure supplement 2B*; *Ródenas et al., 2009*), and on average, ~50% of embryos produced from homozygous *nup53*[tm2886] worms can survive to hatching, as has been shown previously (*Figure 5B*; *Ródenas et al., 2009*). Live imaging of *nup53*[tm2886] one-cell stage embryos revealed that sperm pronuclei nuclear import and nuclear size are similar to *nup53* RNAi-depleted nuclei but reduced compared to control and *ndc1* RNAi-depleted embryos (*Figure 1E-G* and *Figure 5C–E*). RNAi-depletion of *ndc1* strongly enhanced embryonic lethality in the *nup53*[tm2886] strain (*Figure 5B*) - these embryos completely failed to assemble an NE around sperm chromatin (*Figure 5C and D*). Sperm chromatin in *nup53*[tm2886] mutant embryos depleted of *ndc1* remains compacted, and nuclear import is never established (n=7 embryos), even at –100 s relative

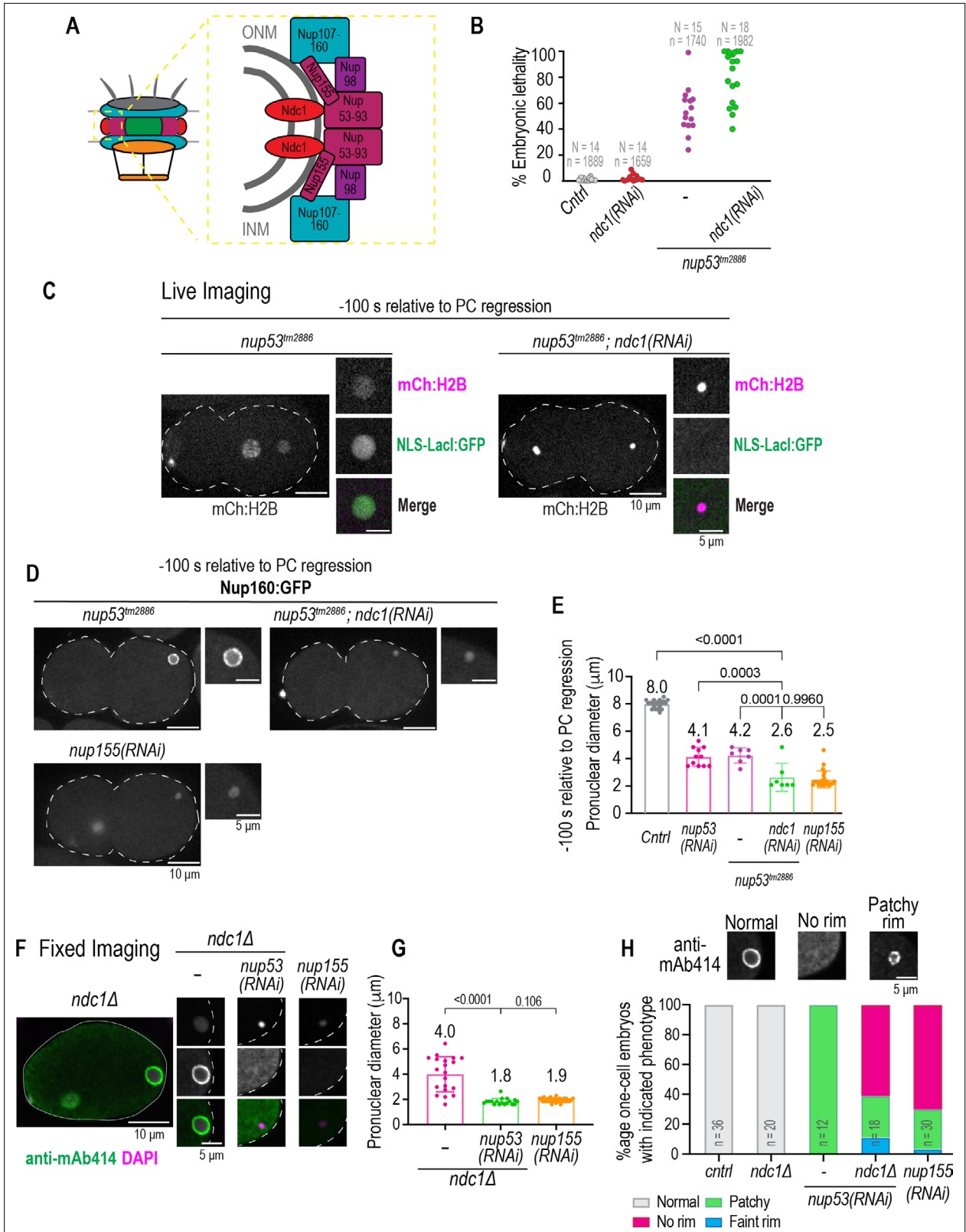

**Figure 5.** Parallel functions for Ndc1 and Nup53 in nuclear assembly. (**A**) Schematic of nuclear pore complex (NPC) (left) and NPC subcomplex organization (right) is shown. (**B**) Plot of percentage embryonic lethality for indicated conditions. N = # of worms. n = # of embryos. (**C**) Confocal overview and magnified images of embryo from a time lapse series of indicated markers for indicated conditions. Scale bars, 10 μm for overview image and 5 μm for magnified images. (**D**) Confocal overview and magnified images of embryo from a time lapse series of Nup160:GFP for indicated

*Figure 5 continued on next page*

*Figure 5 continued*

conditions. Scale bars, 10 µm for overview image and 5 µm for magnified images. (**E**) Plot of pronuclear diameter for indicated conditions at indicated time point. Average ± S.D. is shown. n = # of embryos. *cntrl* n=9, *nup53(RNAi)* n=11, *nup53^{tm2886}* n=7, and *nup53^{tm2886};ndc1(RNAi)* n=7. (**F**) Fixed overview and magnified images of *C. elegans* embryos immunostained for mAb414 and DAPI in indicated conditions. Scale bars, 10 µm for overview image and 5 µm for magnified images. (**G**) Plot of pronuclear diameter for indicated conditions at indicated time point. Average ± S.D. is shown. n = # of embryos. *ndc1Δ* n=20, *ndc1Δ;nup53(RNAi)* n=18, and *nup155(RNAi)* n=30. (**H**) Magnified images of paternal pronucleus from fixed one-cell stage embryos immunostained with mAb414 for indicated conditions (top). Scale bar, 5 µm. Plot of mAb414 appearance surrounding chromatin under indicated conditions (bottom). A two-way ANOVA was used to determine statistical significance between indicated conditions. n = # of embryos. Scale bars, 10 µm for overview image and 5 µm for magnified images.

The online version of this article includes the following source data and figure supplement(s) for figure 5:

**Source data 1.** Excel file containing individual data points related to *Figure 5.*

**Figure supplement 1.** Characterization of *nup53* RNAi depletion and a *nup53* mutant allele.

to PC regression when GFP:NLS accumulates to detectable levels in the sperm-derived pronucleus of *nup53^{tm2886}* embryos (*Figure 5C and E*).

In *nup53^{tm2886}* mutant and *nup53* RNAi-depleted embryos, both Nup160:GFP and mAb414 appear patchy at the nuclear rim in 100% of the nuclei indicating an abnormal distribution of NPCs (*Figure 5D*, see also *Figures 3D and 5H*, *Figure 5—figure supplement 1C*, and *Figure 5—figure supplement 1D*). In contrast, in *nup53^{tm2886}* embryos RNAi depleted of *ndc1*, the Nup160:GFP signal persisted on chromatin (*Figure 5D*), and the nuclear diameter at –100 s prior to PC regression was significantly smaller (*Figure 5E*; average diameter of 2.6 µm versus ~4 µm in *nup53^{tm2886}* or *nup53* [*RNAi*] only conditions, see also *Figure 1D and G* for *ndc1*[*RNAi*]) indicating a failure in assembly of an NE. Thus, Ndc1 is necessary for the recruitment of Nup160:GFP to the nuclear rim and for assembly of nuclei, albeit in a defective manner, when Nup53 dimerization and membrane binding functions are compromised.

Failure to assemble an NE was also observed in fixed *ndc1Δ* embryos RNAi depleted for *nup53* (*Figure 5F-H*). The mAB414 signal around chromatin was normal in *ndc1Δ* embryos and patchy in 100% of fixed one-cell stage embryos RNAi depleted for *nup53* alone (*Figure 5H* and *Figure 5—figure supplement 1C*). The patchy Nup160:GFP (*Figure 5—figure supplement 1D*) and mAB414 (*Figure 5H* and *Figure 5—figure supplement 1C*) signal resulting from RNAi depletion of Nup53 suggest aberrant assembly of NPCs. These NEs still support some nuclear import and expansion (*Figure 5E* and *Figure 1E-G*). In contrast, one-cell stage *ndc1Δ* embryos RNAi depleted for *nup53* contained highly compacted sperm chromatin (*Figure 5F and G*) with little to no mAB414 signal surrounding the chromatin mass (*Figure 5F and H*). Together, these data show that Ndc1 and Nup53 function, at least in part, in parallel pathways to drive NE assembly in early *C. elegans* embryos.

We hypothesized that the redundant function for Ndc1 and Nup53 may be through Nup155, which has been shown to cause severe defects in nuclear formation in *C. elegans* (*Franz et al., 2005*). Indeed, RNAi depletion of Nup155 resulted in complete NE assembly failure that phenocopied loss of both *ndc1* and *nup53* (*Figure 5D–H*). These results suggest that when Ndc1 is absent, Nup53 association with Nup155 is sufficient for NPCs to assemble albeit at a less reduced level.

## Discussion

Our data suggest a model in which Ndc1 functions early during NPC growth to couple incorporation of membranes to stable assembly of the nuclear pore scaffold (*Figure 6A*). In the absence of Ndc1, NPC assembly coupled to membrane incorporation can occur through a parallel pathway that requires Nup53 (*Figure 6A*, middle). In line with this, Nup53 binding to Ndc1 is dispensable for post-mitotic NPC assembly in vitro (*Vollmer et al., 2012*). In addition, Nup53 binds directly to membranes and its overexpression causes membrane proliferation (*Eisenhardt et al., 2014*; *Marelli et al., 2001*; *Vollmer et al., 2012*). Furthermore, in *Drosophila* fat body cells, overexpression of CTDNEP1/CNEP-1 affects the localization of NE-associated Nup53 possibly through disruption of lipid composition at the NE (*Jacquemyn et al., 2021*) also supporting the idea that Nup53 is sensitive to lipid content.

Distinct roles for Ndc1 and Nup53 in membrane incorporation could explain why excess membranes feed nuclear growth in the absence of Ndc1, but not Nup53 (*Figure 6B and C*). One possibility is that Ndc1 contributes to feeding of ER membranes containing preassembled NPC subcomplexes,

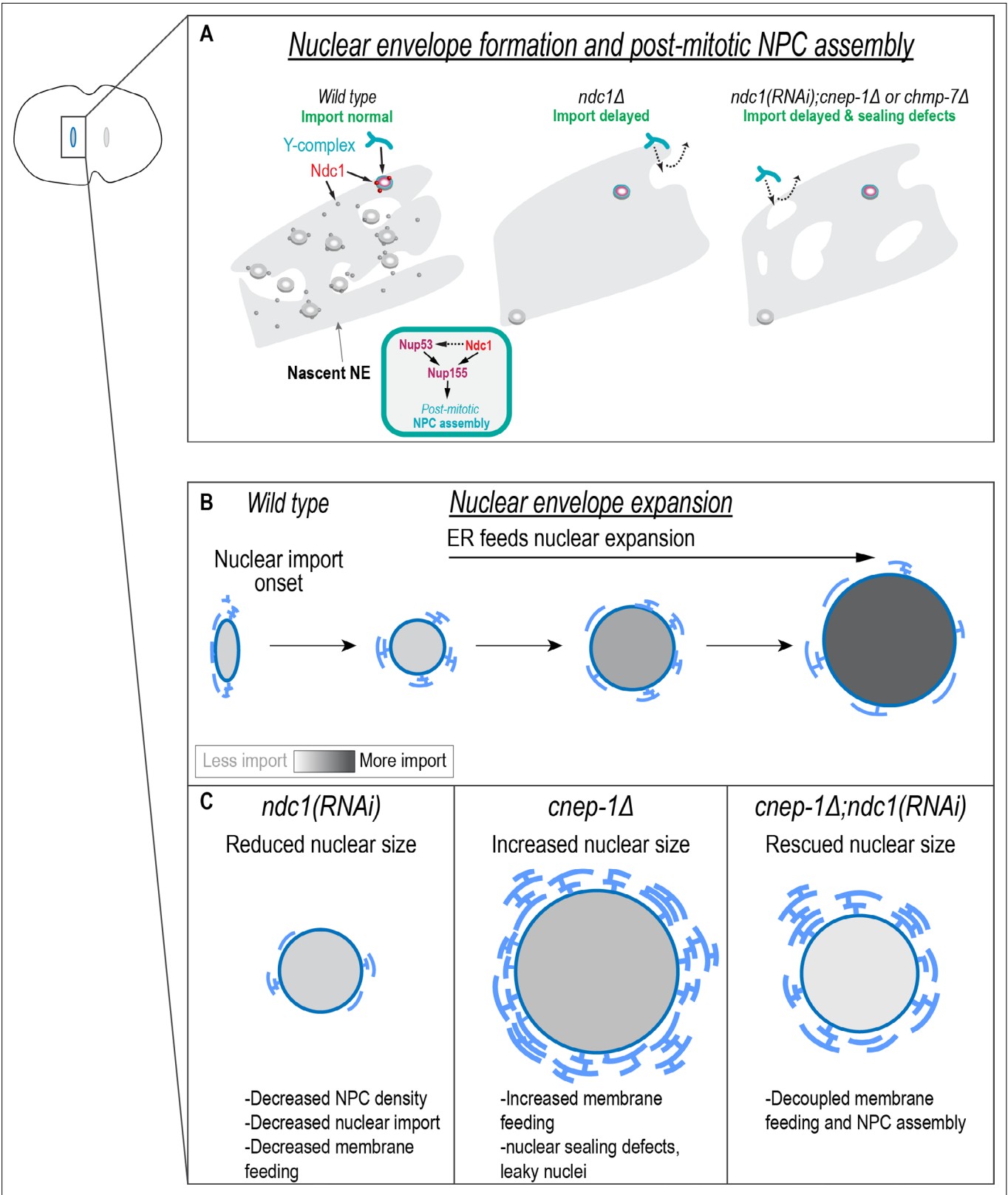

**Figure 6.** Independent requirements for Ndc1 and lipid synthesis in nuclear formation and expansion. (**A**) In the absence of Ndc1, the nuclear envelope (NE) is more continuous, and outer ring scaffold components in the NE are highly dynamic. Some NPCs still assemble and so nuclear transport is eventually established. Ndc1 functions at least in part redundantly to Nup53 in post-mitotic NPC assembly. Both Ndc1 and Nup53 may function through the shared factor Nup155. Loss of Ndc1 in combination with either *cnep-1* or *chmp-7* leads to additional sealing defects (represented by holes in the

*Figure 6 continued on next page*

*Figure 6 continued*

NE) that further delay nuclear formation. (**B**) Endoplasmic reticulum (ER) membranes feed surface area expansion and nuclear import accumulates macromolecules inside the nucleus to increase nuclear volume. (**C**) Independent requirements for Ndc1 and membrane biogenesis promote nuclear growth.

whereas Nup53 is necessary for coupling membrane biogenesis to NPCs assembling directly on chromatin and/or in an intact NE. Indeed, the co-localized puncta of Ndc1 and Nup160 that exist in the cytoplasm of C. elegans embryos may represent annulate lamellae, which form from reserves of precursor NPC condensates to feed nuclear growth as shown in early *Drosophila* embryos (*Hampoelz et al., 2016*; *Hampoelz et al., 2019a*; *Hampoelz et al., 2019b*). The fact that loss of Ndc1 in *C. elegans* embryos also reduces Nup160 localization to cytoplasmic punctate structures suggests that Ndc1 is required for their stability/formation. Interestingly, NPC subcomplexes associate with mitotic ER membranes in mammalian cells to template the formation of mature pores in the reforming NE (*Chou et al., 2021*). Furthermore, annulate lamella also contribute to the rapid repopulation of NPCs in mammalian cells (*Ren et al., 2019*). Thus, there may be a conserved role for Ndc1 in a post-mitotic NPC assembly pathway that involves preassembled NPCs on ER membranes. Future work on the distribution, dynamics, and identity of these Ndc1 containing cytoplasmic puncta in *C. elegans* early embryos and during development will help to define their role in NPC assembly.

Successful NE formation requires Nup155, which assembles proximal to the membrane and links the outer and inner nuclear pore scaffold (*Franz et al., 2005*; *Hampoelz et al., 2019a*; *von Appen et al., 2015*). Although Nup155 can itself bind to membranes (*von Appen et al., 2015*), we found that loss of Nup155 phenocopies loss of both Ndc1 and Nup53. This suggests Nup155 may require that either Ndc1 or Nup53 is present for its recruitment/assembly at the NE (*Figure 6*, top). Stable incorporation of Nup160 also requires Ndc1, and so one possibility is that Ndc1 promotes the self-assembly of the nuclear pore scaffold through Nup155 to drive formation of NPCs. Ndc1 may help to recruit partially assembled nucleoporins that then oligomerize into stable assemblies (see above). Our data showing that Ndc1 is dynamic in growing NEs supports recent data using metabolic labeling in budding yeast showed that Ndc1, unlike other transmembrane nucleoporins, is readily exchanged in the NPC (*Hakhverdyan et al., 2021*; *Onischenko et al., 2020*). However, we also found Ndc1 is highly stable in fully formed nuclei suggesting it serves as an anchor to mature NPCs. Future work is required to determine the mechanism by which Ndc1 drives NPC assembly.

In mammalian cells, Ndc1 slightly delays the onset of nuclear import and nuclear rim formation, consistent with a role in membrane recruitment and post-mitotic NPC assembly, but does not effect GFP:NLS accumulation (*Anderson et al., 2009*). A redundant role for Pom121, which is not present in *C. elegans,* in NPC assembly may account for the differences in nuclear accumulation of GFP:NLS after mitosis in these systems (*Anderson et al., 2009*; *Mansfeld et al., 2006*). We did not detect a delay in bulk membrane recruitment during nuclear formation in *C. elegans* embryos, which could be explained by difficulties in detecting slight delays in membrane recruitment due to the fast speed of nuclear formation as well as limits in the resolution of confocal microscopy to detect defects in membrane incorporation directly at NPC assembly sites.

Deletion of *cnep-1* or *chmp-7* in the absence of Ndc1 further exacerbates the delayed onset of nuclear import resulting from loss of Ndc1 (*Figure 6A*, right). Our prior work in *C. elegans* embryos showed that regulation of the phosphatidic acid phosphatase lipin by CNEP-1 contributes to effective NE sealing by restricting membranes to the surface of chromatin (*Penfield et al., 2020*). Interestingly, in budding yeast, CHMP7 is recruited to phosphatidic acid rich membranes at nuclear membrane herniations suggesting a role for lipid composition in recruiting CHMP7 to the NE (*Thaller et al., 2021*). Together, our data suggest that the onset of nuclear cargo accumulation is likely limited by the number of assembled NPCs and the presence of holes that allow passive diffusion between the nucleoplasm and cytoplasm, which require closure through restricting lipid synthesis and ESCRT-mediated mechanisms (*Figure 6*).

Membrane biogenesis coupled to NPC biogenesis has been proposed to drive faster NE expansion by both maintaining nuclear transport rates and supporting faster surface area expansion (*Kume et al., 2019*; *Kume et al., 2017*). In budding yeast, nuclear extrusions resulting from increased nuclear membrane biogenesis through deletion of *nem1* (*cnep-1/CTDNEP1* in yeast) or its binding partner *spo7* contain NPCs suggesting that NPC assembly is coupled to membrane biogenesis under these

conditions (*Jaspersen and Ghosh, 2012*; *Santos-Rosa et al., 2005*; *Siniossoglou, 2013*; *Siniossoglou et al., 1998*). Our data show that faster rates of nuclear expansion driven by excess membrane biogenesis do not require NPC assembly via Ndc1 nor maintenance of normal rates of nuclear import (*Figure 6C*). Why then is the rate of nuclear surface area expansion limited when the ER normally contains a large pool of available membranes? The amount of peripheral ER membranes determines the size of the nucleus in early sea urchin embryos (*Hara and Merten, 2015*; *Mukherjee et al., 2019*). *cnep-1* mutants contain excess ER membranes sheets proximal to the NE (*Bahmanyar et al., 2014*) supporting the possibility that a specific ER membrane pool controlled by CNEP-1 more readily feeds nuclear expansion (*Figure 6C*). Additionally, CNEP-1/CTDNEP1 is enriched at the NE, and so membrane biogenesis may occur locally at the NE and couple to NPC assembly via a pathway that involves Nup53 (*Bahmanyar et al., 2014*; *Jacquemyn et al., 2021*). Finally, local lipid composition may also be regulated by this pathway to direct membrane flow and NPC assembly.

Lipid synthesis is a universal mechanism that controls nuclear size and shape (*Barger et al., 2022*; *Grillet et al., 2016*; *Kume et al., 2017*; *Merta et al., 2021*; *Santos-Rosa et al., 2005*), and future work is needed to understand the direct and indirect relationships between ER/NE membrane biogenesis, NPC assembly pathways, and nuclear transport rates to regulate NPC density, nuclear expansion, and nuclear size.

## Resource availability

### Lead contact
Further information and requests for reagents or resources should be directed to Shirin Bahmanyar ( shirin.bahmanyar@yale.edu).

### Materials availability
*C. elegans* strains and plasmids from this study are available from the Lead Contact upon request.

## Experimental model and subject details

### Strain maintenance and generation
The *C. elegans* strains used in this study are listed in the key resource table. Strains were maintained on nematode growth media plates that were seeded with OP-50 *Escherichia coli*. Strains were grown at three possible temperatures: 15, 20, and 25°C.

### CRISPR-Cas9 deletion strain
The deletion for *ndc1* (B0240.4) was generated using two CRISPR guides 'crRNA', which were obtained using IDT's custom CRISPR guide algorithm (see *Figure 1—figure supplement 1A* and key resource table for sequences). 1 µl of the purified crRNAs were annealed to 1 µl of trans-activating crRNA by incubating RNAs at 95° C for 2 min in individual PCR tubes. A co-CRISPR crRNA was used for *dpy-10*. An injection mix with the following components and concentrations was set up and spun down for 30 min at 4°C: *ndc1* guide 1 (11.7 µM), *ndc1* guide 2 (11.7 µM), purified Cas9 protein (qb3 Berkeley, 14.7 µM), *dpy-10* guide (3.7 µM), and a *dpy-10* repair template (29 ng/mL) (*Arribere et al., 2014*; *Paix et al., 2015*).

The RNA-protein mix was then injected into the gonads of N2 adult worms, which were then allowed to recover for 3 days. After this recovery, F1 progeny was screened for a roller phenotype, and 10–20 F1s were singled out to individual plates. These roller worms were allowed to produce progeny, which were then genotyped by PCR to detect the presence of the *npp-22/ndc1* deletion allele. The deletion strain was outcrossed four times to N2 (ancestral strain) worms before use and characterization.

### CRISPR-Cas9 with SEC repair template for endogenous tagging
To endogenously tag *ndc1* with mNeonGreen and mRuby, a self-excising cassette (SEC) repair template approach was utilized. 950 bp homology arms from the five- and three-prime end of *ndc1*'s stop codon were cloned into an SEC vector. This plasmid was co-injected with a Cas9+ndc1 guide plasmid into the gonads of adult N2 worms. Worms were rescued to individual plates and allowed

to recover for 3 days at 20°C. Plates were screened for roller worms and positive plates were treated with approximately 200–300 µl of hygromycin B (20 mg/mL) and were allowed to recover for 4–5 days. Plates that had surviving rolling worms were screened for mNeonGreen or mRuby signal by microscopy. Finally, worms were outcrossed four times to N2 worms before crossing to other fluorescent markers.

# Materials and methods

## Key resources table

| Reagent type (species) or resource | Designation | Source or reference | Identifiers | Additional information |
|---|---|---|---|---|
| Antibody | mouse monoclonal α-alpha-tubulin | Millipore Sigma | Cat#05–829; RRID: AB_310035 | 1 µg/mL |
| Antibody | mouse monoclonal mAb414 | Biolegend | Cat# 902907; RRID: AB_2734672 | WB: 1 µg/mL IF:2.5 µg/mL |
| Antibody | mouse monoclonal mAb414 | Millipore-Sigma | MABS1267 | WB: 1 µg/mL IF: 2.5 µg/mL |
| Antibody | rabbit polyclonal α-NPP-5/Nup107 | *Ródenas et al., 2009* | N/A | WB: 1:300 IF: 1:300 |
| Antibody | rabbit polyclonal α-NPP-10N/Nup98 | *Ródenas et al., 2009* | N/A | WB: 1:500 |
| Antibody | rabbit polyclonal α-NPP-10C/Nup96 | *Ródenas et al., 2009* | N/A | WB: 1:500 |
| Antibody | rabbit polyclonal α-NPP-19/Nup53 | *Ródenas et al., 2009* | N/A | WB: 1:1,000 IF: 1:300 |
| Antibody | rabbit polyclonal α-MEL-28/Elys | *Ródenas et al., 2009* | N/A | IF: 1:500 |
| Antibody | rabbit polyclonal α-IMA-3 | *Geles and Adam, 2001* | N/A | WB: 1:400 |
| Antibody | rabbit polyclonal α-LMN1 | *Penfield et al., 2018* | N/A | IF: 1 µg/mL |
| Antibody | Rhodamine RedX donkey polyclonal α rabbit IgG | Jackson Immuno | Cat#711-295-152; RRID: AB_2340613 | IF: 1:200 |
| Antibody | FITC goat polyclonal α mouse IgG | Jackson Immuno | Cat#115-095-146; RRID: AB_2338599 | IF: 1:200 |
| Antibody | goat polyclonal α mouse IgG-HRP | Thermo Fisher | Cat#31430; RRID: AB_228307 | WB: 1:7,000 |
| Antibody | goat polyclonal α rabbit IgG-HRP | Thermo Fisher | Cat#31460; RRID: AB_228341 | WB: 1:5,000 |
| Commercial assay or kit | Clarity Max Western ECL Substrate | BIO-Rad | 1705060 S | |
| Commercial assay or kit | MEGAscript T3 Transcription Kit | Invitrogen | Cat# AM1338 | |
| Commercial assay or kit | MEGAscript T7 Transcription Kit | Invitrogen | Cat# AM1334 | |
| Strain and strain background (*Caenorhabditis elegans*) | *C. elegans*: Strain N2: wildtype (ancestral) | *Caenorhabditis* Genetics Center | N2 | |
| Strain and strain background (*C. elegans*) | *C. elegans*: Strain OD997: *unc-119(ed3)III; ltSi231[pNH16; Pmel-28::GFP-mel-28; cb-unc- 119(+)]III;; ltIs37[pAA64; pie-1/ mCherry::his-58; unc-119 (+)] IV* | *Hattersley et al., 2016* | OD997 | |

*Continued on next page*

Continued

| Reagent type (species) or resource | Designation | Source or reference | Identifiers | Additional information |
|---|---|---|---|---|
| Strain and strain background (C. elegans) | C. elegans: Strain OD999: unc-119(ed3)III; ltSi245[pNH42; Pnpp-18::GFP-npp-18; cb-unc- 119(+)]II; ltIs37[pAA64; pie-1/ mCherry::his-58; unc-119 (+)] IV | Hattersley et al., 2016 | OD999 | |
| Strain and strain background (C. elegans) | C. elegans: Strain OD1496: unc-119(ed3)III; ltSi464[pNH103; Pmex-5::npp-6::GFP::tbb-2:3'UTR; cbunc-119(+)]I; ltIs37[pAA64; pie-1/mCherry::his-58; unc- 119 (+)] IV | Hattersley et al., 2016 | OD1496 | |
| Strain and strain background (C. elegans) | C. elegans: Strain OD1498: unc-119(ed3)III; ltSi465[pNH104; Pmex-5::npp-15::GFP::tbb-2:3'UTR; cb-unc-119(+)]I; ltIs37[pAA64; pie-1/mCherry::his-58; unc-119 (+)] IV | Hattersley et al., 2016 | OD1498 | |
| Strain and strain background (C. elegans) | C. elegans: Strain OD1499: unc-119(ed3)III; ltSi463[pNH102; Pmex-5::npp2::GFP::tbb-2:3'UTR; cb- unc-119(+)]I; ltIs37[pAA64; pie-1/mCherry::his-58; unc-119 (+)] IV | Hattersley et al., 2016 | OD1499 | |
| Strain and strain background (C. elegans) | C. elegans: Strain OD2400: ltSi896[pNH152; Pgsp-2::GSP-2::GFP; cb-unc-119(+)]I; ltIs37[pAA64; pie-1/mCherry::his-58; unc-119 (+)] IV | Hattersley et al., 2016 | OD2400 | |
| Strain and strain background (C. elegans) | C. elegans: Strain SBW32: unc-119(ed3) III; ltIs24 [pAZ132; pie-1/ GFP::tba-2; unc-119 (+)]; ltIs37 [(pAA64) pie-1p::mCherry::his-58+unc-119(+)] IV | Penfield et al., 2018 | SBW32 | |
| Strain and strain background (C. elegans) | C. elegans: Strain SBW47:unc-119(ed3) III; ltIs37 [pie-1/ mCherry::his-58; unc-119 (+)] IV; ltIs75 [Ppie-1/GFP::TEV-Stag::LacI +unc-119(+)]. | Penfield et al., 2020 | SBW47 | |
| Strain and strain background (C. elegans) | C. elegans: Strain SBW56: npp-22(ndc1) (tm1845/nT1) V 7 x outcrossed | This study | SBW56 | Related to Figure 1—figure supplement 1D and E |
| Strain and strain background (C. elegans) | C. elegans: Strain SBW65: scpl-2(tm4369)II;unc-119(ed3) III; ltIs37 [pAA64; pie-1/mCHERRY::his-58; unc-119 (+)] IV;ltIs75 [(pSK5) pie-1::GFP::TEV-STag::LacI +unc-119(+)]. | Penfield et al., 2020 | SBW65 | |
| Strain and strain background (C. elegans) | C. elegans: Strain SBW79: chmp-7 (T24B8.2) deletion II; unc-119(ed3) III; ltIs37 [pAA64; pie-1/mCHERRY::his-58; unc-119 (+)] IV;ltIs75 [(pSK5) pie-1::GFP::TEV-STag::LacI +unc-119(+)]. | Penfield et al., 2020 | SBW79 | |
| Strain and strain background (C. elegans) | C. elegans: Strain SBW83: npp-22(ndc1) (sbw4) V, 4 x outcrossed | This study | SBW83 | Related to Figure 1B and C, Figure 1—figure supplement 1A-B |
| Strain and strain background (C. elegans) | C. elegans: Strain SBW84: unc-119(ed3) III; bqSi242 [lem-2p::lem-2::mCherry +unc-119(+)] IV; ltIs75 [(pSK5) pie-1::GFP::TEV-STag::LacI +unc-119(+)]. | This study | SBW84 | Related to Figure 2C and D |
| Strain and strain background (C. elegans) | C. elegans: Strain SBW191: npp-19(tm2886) 6× outcrossed | This study | SBW191 | Related to Figure 5B, Figure 5—figure supplement 1B |
| Strain and strain background (C. elegans) | C. elegans: Strain SBW244: ndc1::mNeonGreen (sbw8)–4× outcrossed | This study | SBW244 | Related to Figure 3E and Figure 3—figure supplement 1G |
| Strain and strain background (C. elegans) | C. elegans: Strain SBW245: unc-119(ed3) III; ltIs37 [pAA64; pie-1/mCHERRY::his-58; unc-119 (+)] IV; ndc1::mNEON (sbw8)–4× outcrossed | This study | SBW245 | Related to Figure 2—figure supplement 1B |

*Continued*

| Reagent type (species) or resource | Designation | Source or reference | Identifiers | Additional information |
|---|---|---|---|---|
| Strain and strain background (*C. elegans*) | *C. elegans:* Strain SBW252: *unc-119(ed3) III; ltIs 76 [pAA178; pie-1/mCherry:SP-12; unc-119 (+)]; ndc1::mNEON (sbw8)–4× outcrossed* | This study | SBW252 | Related to ***Figure 2— figure supplement 1C*** |
| Strain and strain background (*C. elegans*) | *C. elegans:* Strain SBW254:*bqSi242 [lem-2p::lem-2::mCherry +unc-119(+)] IV.; ndc1::mNeonGreen (sbw8)–4× outcrossed* | This study | SBW254 | Related to ***Figure 2B*** |
| Strain and strain background (*C. elegans*) | *C. elegans:* Strain SBW260: npp-19(tm2886) outcross 6 x; (unc-119(ed3)III; ltSi464[pNH103; Pmex-5::npp6::GFP::tbb-2:3'UTR; cb-unc-119(+)]I) | This Study | SBW260 | Related to ***Figure 5D*** |
| Strain and strain background (*C. elegans*) | *C. elegans:* Strain SBW266: npp-19(tm2886) outcross 6 x; unc-119(ed3) III; ltIs37 [pAA64; pie-1/mCHERRY::his-58; unc-119 (+)] IV; ltIs75 [(pSK5) pie-1::GFP::TEV-STag::LacI +unc-119(+)]. | This Study | SBW266 | Related to ***Figure 5C*** |
| Strain and strain background (*C. elegans*) | *C. elegans:* Strain SBW293: *(unc-119(ed3)III; ltSi464[pNH103; Pmex-5::npp6::GFP::tbb-2 3'UTR; cb-unc-119(+)]I) V; ndc1::mRuby (sbw14)–4 x outcrossed* | This study | SBW293 | Related to ***Figure 3— figure supplement 1C*** |
| Sequence-based reagent | T3 primer for dsRNA targeting *npp-22(ndc1)* Forward: AATT AACCCTCACTAAAGGCCCGCCTCCATATACAGTTC | This Study | N/A | Used to generate dsRNA for RNAi knock down of *ndc1* |
| Sequence-based reagent | T7 primer for dsRNA targeting *npp-22(ndc1)* Reverse: TAAT ACGACTCACTATAGGTGTCAATGGCTGCAATGAGT | This Study | N/A | Used to generate dsRNA for RNAi knock down of *ndc1* |
| Sequence-based reagent | T3 primer for dsRNA targeting *npp-7 (nup153)* Forward: AATT AACCCTCACTAAAGGGTTCCTGCCACAATTCCAGT | This Study | N/A | Used to generate dsRNA for RNAi knock down of *npp-7/nup153* |
| Sequence-based reagent | T7 primer for dsRNA targeting *npp-7 (nup153)* Reverse: TAAT ACGACTCACTATAGGCTTGTAGACGATGCAGCACC | This Study | N/A | Used to generate dsRNA for RNAi knock down of *npp-7/nup153* |
| Sequence-based reagent | T3 primer for dsRNA targeting *npp-19 (nup53)* Forward: AATT AACCCTCACTAAAGGCACCACCTCTTCGATCTCTTC | This Study | N/A | Used to generate dsRNA for RNAi knock down of *npp-19/nup53* |
| Sequence-based reagent | T7 primer for dsRNA targeting npp-19 (nup53) Reverse: TAAT ACGACTCACTATAGGTTTGTGCACTGAACGACTCC | This Study | N/A | Used to generate dsRNA for RNAi knock down of *npp-19/nup53* |
| Sequence-based reagent | T7 primer for dsRNA targeting *npp-8 (nup155)* Forward: TAAT ACGACTCACTATAGGGATTTGGCGTTTTTCGACTC | This Study | N/A | Used to generate dsRNA for RNAi knock down of *npp-8/nup155* |
| Sequence-based reagent | T7 primer for dsRNA targeting npp-8 (nup155) Reverse: TAAT ACGACTCACTATAGGCACGAAATCAAAGACCGGAT | This Study | N/A | Used to generate dsRNA for RNAi knock down of *npp-8/nup155* |

*Continued*

| Reagent type (species) or resource | Designation | Source or reference | Identifiers | Additional information |
|---|---|---|---|---|
| Sequence-based reagent | T7 primer for dsRNA targeting *npp-10 (nup96/98)* Forward: TAATACGACTCACTATAGGAGTTCATTGTTCGGTGGAGG | This Study | N/A | Used to generate dsRNA for RNAi knock down of *npp-10/nup96/98* |
| Sequence-based reagent | T7 primer for dsRNA targeting npp-10 (nup96/98) Reverse: TAAT ACGACTCACTATAGGATTGGAACCAAAAATGCTGC | This Study | N/A | Used to generate dsRNA for RNAi knock down of *npp-10/nup96/98* |
| Sequence-based reagent | *npp-22/ndc1* (B0240.4) start of gene CRISPR guide: AGTGAATTAGAGTTCCAAAC | This Study | N/A | Related to *Figure 1—figure supplement 1A* |
| Sequence-based reagent | *npp-22/ndc1* (B0240.4) end of gene CRISPR guide: AGGAACACTCACGAACCATT | This Study | N/A | Related to *Figure 1—figure supplement 1A* |
| Sequence-based reagent | *npp-22/ndc1* quantitative PCR (qPCR) primer forward: AGCTGTTTCCTTGCCTTGTG | This Study | N/A | Related to *Figure 1—figure supplement 1B* |
| Sequence-based reagent | *npp-22/ndc1* qPCR primer reverse: TCTTGGCATCAGGAGAGCAT | This Study | N/A | Related to *Figure 1—figure supplement 1B* |
| Sequence-based reagent | *pmp-3 (house-keeping gene)* qPCR primer forward: GGTCATCGGTATTCGCTGAA | *Chauve et al., 2021* | N/A | |
| Sequence-based reagent | *pmp-3 (house-keeping gene)* qPCR primer reverse: *GAGGCTGTGTCAATGTCGTG* | *Chauve et al., 2021* | N/A | |
| Recombinant DNA reagent | Plasmid: PDD122, CRISPR-Cas9 | *Hastie et al., 2019* | N/A | |
| Recombinant DNA reagent | Plasmid: pSB446; CRISPR npp-22/*ndc1* guide inserted using gibson-assembly | This Study | N/A | Related to *Figure 2—figure supplement 1A* |
| Recombinant DNA reagent | Plasmid: pBS-LL-mNG | *Hastie et al., 2019* | N/A | |
| Recombinant DNA reagent | Plasmid: pSB448; 1 kb homology arms of *npp-22/ndc1* C-term inserted into pBS-LL-mNG using gibson-assembly | This Study | N/A | Related to *Figure 2—figure supplement 1A* |
| Recombinant DNA reagent | Plasmid: LL-mRuby | This Study | N/A | Related to *Figure 2—figure supplement 1A* |
| Recombinant DNA reagent | Plasmid: pSB603; 1 kb homology arms of *npp-22/ndc1* C-term inserted into LL-mRuby using gibson-assembly | This Study | N/A | Related to *Figure 2—figure supplement 1A* |

*Continued on next page*

| Reagent type (species) or resource | Designation | Source or reference | Identifiers | Additional information |
|---|---|---|---|---|
| Software and algorithm | FIJI (ImageJ) | NIH | https://imagej.net/Fiji RRID: SCR_002285 | |
| Software and algorithm | IMOD Version 4.11 | University of Colorado | https://bio3d.colorado.edu/imod/ | |
| Software and algorithm | GraphPad Prism 8/9 | GraphPad | N/A | |
| Software and algorithm | R Studio Version 1.2.5033 | R Studio, INC | https://www.rstudio.com | |
| Software and algorithm | MATLAB | MathWorks, INC | https://www.mathworks.com/products/matlab.html | |
| Software and algorithm | CytoShow | CytoShow | http://www.cytoshow.org/ | |

## RNA interference

Primers were designed to amplify a 200–1000 bp region within a gene of interest, see key resource table for list of primers used. Primers were designed to amplify from within a single exon whenever possible. N2 gDNA or cDNA was used as a template for PCR, which in turn was purified and used in T3/T7 reverse transcription reactions (MEGAscript, Life Technologies). The synthesized RNAs were purified using phenol-chloroform and resuspended in 1× soaking buffer (32.7 mM $Na_2HPO_4$, 16.5 mM $KH_2PO_4$, 6.3 mM NaCl, and 14.2 mM $NH_4Cl$). RNA reactions were annealed at 68°C for 10 min followed by 37°C for 30 min. dsRNAs were brought to a final concentration of ~2000 ng/µl whenever possible, and 2 µl aliquots of the dsRNA were stored until use at –80°C. For each experiment, prior to injection, a fresh aliquot was diluted to ~1000 ng/µl and centrifuged at 13,000 rpm for 30 min at 4°C. 0.35 µl of the diluted dsRNA was loaded into the back of hand pulled capillary needles and injected into the gut of L4 worms. Worms were rescued to plates seeded with OP-50 and allowed to recover for ~24 hr prior to imaging or lethality analysis.

## Lethality quantifications

L4 worms were injected with indicated dsRNA and allowed to recover for 24 hr at 20°C. 24 hr post-injection worms were then singled out and allowed to self-fertilize for an additional 24 hr. Worms were transferred to another plate for a final 24 hr, then disposed of. Plates corresponding to 24–48 hr and 48–72 hr post-injection were then scored for hatched larvae and unhatched embryos. Prior to counting, embryos were given 24 hr to hatch. The total number of embryos and larvae were combined for each time window to calculate embryonic lethality and brood size. A similar approach was used for non-injected control worms and worms containing deletion alleles.

## Immunoblots

### Generation of whole worm lysate

For a given condition, a microcentrifuge tube was filled with 30 µl of M9 Buffer, and the fill line was marked with a black marker. For each condition, 35 adult worms were then placed in the microcentrifuge tube and washed three times with M9+0.1% Triton. After the final wash, tubes were brought up to a final volume of 30 µl. Then, 10 µl of 4× sample buffer was added and the tubes were mixed. The samples were then sonicated at 70°C for 15 min, followed by incubation for 5 min at 95°C. Samples were re-sonicated at 70°C for an additional 15 min. Worm lysates were stored at –20°C until they were run on an SDS-PAGE protein gel.

### Protein gel electrophoresis and antibody probing

For all protein gels, homemade 8–10% SDS-PAGE were used (blot in *Figure 3—figure supplement 2D* was a 3–8% Bio-Rad Tris-Acetate gel). Worm lysates were re-boiled at 95°C for 5 min, and then 20 µl (~17.5 worms) were loaded into each lane. The protein gel was then run at 80 V for 15 min to fully collapse samples. The protein gel was then run at 120 V for approximately 90 min, or until

the sample buffer reached the bottom of the gel. Protein samples were then transferred overnight (16 hr at 4°C) to an nitrocellulose membrane at 100 mA. Membranes were blocked in fresh 5% milk in TBST for 1 hr at room temperature; membranes were then cut based on size and incubated overnight with the following primary antibodies: 1 µg/mL mouse α-alpha-tubulin (EMD Millipore), 1 µg/mL mouse mAb414 (Biolegend), rabbit α-NPP-5/Nup107 (1:300), rabbit α-NPP-10N/Nup98 (1:500), rabbit α-NPP-10C/Nup96 (1:500), and rabbit α-NPP-19/Nup53 (1:1000) rabbit α-Ima-3 (1:400).

Membranes were then rinsed three times quickly with TBST followed by four 5-min washes. Membranes were then incubated with appropriate secondary antibodies for 1 hr at room temperature. Secondary antibodies were diluted 1:5000 for horseradish peroxidase (HRP)-conjugated goat-anti-rabbit and 1:7000 for HRP-conjugated goat-anti-mouse (Thermo Fischer Scientific). Membranes were rinsed and washed as described above. Prior to image acquisition, membranes were incubated with Clarity Max Western ECL Substrate (BIO-RAD) for 4 min and then excess reagent was removed.

## Immunofluorescence
### Slide preparation
Microscope slides (Fisher Scientific Premium Microscope Slides Superfrost) were coated with 0.1% polylysine and dried on a low temperature heat block. Slides were then baked at 95°C for 30 min. Slides were used the same day that they were baked.

### Fixation and primary antibody incubation
A range of 15–20 adult worms were picked into a 4 µl drop of ddH$_2$O and covered with a standard 18 × 18 mm coverslip. Embryos were pushed out of the adult worms by pressing down on the corners of the coverslip with a pipet tip. To crack the eggshell and permeabilize the embryos, slides were placed in liquid nitrogen for ~5 min. Coverslips were quickly removed by using a razor blade to pop off the coverslip. Slides were then fixed in pre-chilled 100% methanol at –20°C for 20 min. Following fixation slides were washed two times in 1× PBS + 0.2% Tween 20 (PBST) at room temperature for 10 min each. After the second wash, samples were blocked with 50 µl of 1% BSA in PBST per slide in a humid chamber for 1 hr at 20°C. Slides were then incubated overnight at 4°C with primary antibodies diluted in PBST (45 µl per slide; rabbit α-LMN1, 1 µg/mL, mouse mAb414 2.5 µg/mL, rabbit α-NPP-5/Nup107 [1:300], rabbit α-NPP-19/Nup53 [1:300], and rabbit α-MEL-28/Elys [1:500]). For the immunofluorescence experiment in *Figure 3—figure supplement 2C* (mAb414 in control and *ndc1Δ* embryos) the above fixation protocol was amended to include a 15 min fixation at room temperature with 4% paraformaldehyde prior to the methanol fixation.

### Secondary antibody incubation and DAPI +Hoechst staining
After overnight primary antibody incubation, slides were washed two times in (50 µl per slide) PBST at room temperature for 10 min each. Following the second wash, slides were incubated at 20°C for 1 hr in the dark with secondary antibodies diluted in PBST (45 µl per slide, anti-rabbit Cy3/Rhodamine, 1:200; anti-mouse FITC, 1:200; [Jackson Immunoresearch]). Slides were again washed two times in (50 µl per slide) PBST at room temperature for 10 min each in the dark. Samples were stained with 1 µg/mL Hoechst (diluted from a 1 mg/mL stock in H$_2$O). Slides were washed two final times in (50 µl per slide) PBST at room temperature for 10 min each in the dark. Finally, mounting media (Molecular Probes ProLong Gold Antifade Reagent with DAPI) was added to each sample. Coverslips were gently placed onto the slides and adhered with clear nail polish. Slides were allowed to dry at 20°C and then stored at –20°C until they were imaged.

## Image acquisition
### Live microscopy
First, 2% agarose imaging pads were made using molten agarose (95°C) on a glass slide. Adult hermaphrodites were then dissected on glass slides, and the embryos were transferred to 2% agarose imaging pads using a mouth pipette at 20°C. Embryos were organized using an eyelash tool to group similar stage embryos. Images were acquired every 20 s. Five z-slices were taken for each time point with 2 µm steps. Images were acquired on an inverted Nikon Ti microscope equipped with a confocal scanner unit (CSU-XI, Yokogawa). Two solid state lasers (100 mW 488 nm and 50 mW 561 nm) were

used in conjunction with a 60× objective lens (Å~1.4 NA Plan Apo). Images were recorded with a high-resolution ORCA R-3 Digital CCD Camera (Hamamatsu).

### Fluorescence recovery after photobleaching

First, 2% agarose imaging pads were made using molten agarose (95°C) on a glass slide. Adult hermaphrodites were then dissected on glass slides, and two-cell stage embryos were transferred to 2% agarose imaging pads using a mouth pipette at 20°C. Embryos were organized using an eyelash tool to group similar stage embryos. A stimulation ROI was then drawn on a region of the NE of an AB nucleus (two-cell stage embryo). Three images were taken prior to stimulation/bleaching of the ROI by a 100 mW, 405 nm laser. Images were then acquired for the remainder of the cell cycle, until NE breakdown. Images were taken every 10 s with one z-slice taken per time point.

### Fixed microscopy

Immunofluorescent images were acquired on an inverted Nikon Ti Eclipse microscope. This microscope was equipped with solid state 100 mW 405, 488, 514, 594, 561, 594, and 640 nm lasers, a Yokogawa CSU-W1 confocal scanner unit, a 60 × 1.4 NA Plan Apo objective lens, and a prime BSI sCMOS camera.

### Microdevice imaging

Fabricated microdevices were initially cleaned with ddH$_2$O and then filled with 0.75× egg salts (*Carvalho et al., 2011*). Three worms were placed in the microdevice and dissected to release early embryos. One-cell stage embryos were pushed into the imaging wells using an eyelash tool. Embryos floated to the bottom of the well and then were imaged every 10 s with 11 slices taken at 2 μm steps. Related to *Figure 5A–E*.

### diSPIM imaging

Data related to *Figure 4A and B* and *Figure 4—video 1* were collected using dual-view inverted light sheet microscopy (diSPIM; *Kumar et al., 2014*). Embryos were imaged in 0.75× egg salts and images were acquired every 20 s with a Z step of.5 μm. Two 40×, 0.8 NA, water dipping objectives were used in conjunction with a 50 mW, 488 nm laser (Newport, PC14584) and a Hamamatsu Flash 4.0 sCMOS camera. The two image stacks were deconvolved using CytoShow (CytoSHOW can be downloaded from http://www.cytoshow.org/) and yielded images with isotropic resolution (0.1625 μm in X, Y, and Z).

## Quantitative PCR (qPCR)

For each strain (10–15 medium plates) unsynchronized populations of adult worms were washed with 1× M9 buffer into 15 mL conical tubes. Worms were washed with 5 mL of M9, three times. After the final wash, the buffer was removed from the worm pellet and the samples were frozen at –80°C. Next an approximate 100 μl worm pellet was ground using a motorized micro pestle. 1 mL of Trizol was then added to the ground worm pellet, and the mixture was vortexed for 15 min at room temperature. Following vortexing, 200 μl of chloroform was added to the sample. The trizol-chloroform mixture was then briefly vortexed (15 s) followed by an incubation at room temperature for 5 min. Next the solution was spun for 15 min at 4°C (12,000 rpm). To extract the total RNA, the upper aqueous layer was removed (approximately 500 μl) and transferred to a new microcentrifuge tube. 500 μl of isopropanol was added to the tube, and it was inverted six times.

To precipitate the RNA, the solution was incubated at room temperature for 10 min followed by a 10 min spin 4°C (12,000 rpm). The supernatant was discarded, and the pellet was washed in 1 mL of 75% ethanol by inverting once and vortexing for 10 s. To pellet the RNA, the solution was spun for 5 min at 4°C (7500 rpm), the supernatant was removed, and the pellet was air-dried at room temperature for 5 min. The pellet was resuspended in 87 μl of RNase-free water and incubated at 37°C for 15 min. After incubation, the isolated RNA was mixed with 10 μl NEB Buffer 4, 1 μl of 50 μM CaCl$_2$, and 2 μl Ambion Turbo DNase and incubated at room temperature for 15 min. The RNA/DNase reaction was cleaned with a phenol-chloroform extraction and eluted in 20 μl of RNase-free water. 500 ng

of total RNA was reverse transcribed using Invitrogen Super-Script II Reverse Transcriptase kit. The cDNA was stored at –20°C.

For qPCR reactions, cDNA (~1200 ng/μl) was diluted fivefold and prepared for amplification using a BioRad SYBR Green Supermix Kit with the primers for nucleoporin genes listed in key resource table as well as *pmp-3* primers from *Chauve et al., 2021*. For each biological replicate (two wild type and two *ndc1Δ*), four technical replicates were performed. Reactions were loaded into a 384-well plate for amplification. The reactions were amplified, and *Ct* values were measured using a BioRad CFX 384 qPCR machine. An annealing temperature of 55°C was used for all genes. After amplification, *Ct* values for each nucleoporin gene were normalized to *hxk-2*, and the fold change relative to the control was calculated using the $2^{-(\Delta\Delta Ct)}$ method (BioRad).

## Transmission electron microscopy

### Sample preparation

Wild-type N2 and SBW83 *C. elegans* hermaphrodites were dissected in M9 buffer, and single embryos early in mitosis were selected and transferred to cellulose capillary tubes (Leica Microsystems, Vienna, Austria) with an inner diameter of 200 μm. The embryos were observed with a stereomicroscope until cleavage furrow ingression in late anaphase and then immediately cryo-immobilized using a LEICA ICE high-pressure freezer (Leica Microsystems, Vienna, Austria). Freeze substitution was performed over 3 days at –90°C in anhydrous acetone containing 1% $OsO_4$ and 0.1% uranyl acetate using an automatic freeze substitution machine (EM AFS, Leica Microsystems, Vienna, Austria). Epon/Araldite infiltrated samples were flat embedded in a thin layer of resin, polymerized for 2 days at 60°C, and selected by light microscopy for re-mounting on dummy blocks. Serial semi-thick sections (200 nm) were cut using an Leica Ultracut S Microtome (Leica Microsystems, Vienna, Austria). Sections were collected on Pioloform-coated copper slot grids and post-stained with 2% uranyl acetate in 70% methanol followed by Reynold's lead citrate.

### Data acquisition by electron tomography

Colloidal gold particles (15 nm; Sigma-Aldrich) were attached to both sides of semi-thick sections collected on copper slot grids to serve as fiducial markers for subsequent image alignment. For dual-axis electron tomography, series of tilted views were recorded using an F20 electron microscopy (Thermo-Fisher, formerly FEI) operating at 200 kV at magnifications ranging from 5000× to 6500× and recorded on a Gatan US4000 (4000 px × 4000 px) CCD or a Teitz TVIPS XF416 camera. Images were captured every 1.0° over a ±60° range.

### 3D reconstruction and automatic segmentation of MTs

We used the IMOD software package (http://bio3d.colourado.edu/imod), which contains all of the programs needed for calculating electron tomograms. For image processing the tilted views were aligned using the positions of the colloidal gold particles as fiducial markers. Tomograms were computed for each tilt axis using the R-weighted back-projection algorithm.

### NE and NPC segmentation and measurement

The IMOD software package was used to segment the NE and NPCs in control and *ndc1Δ* tomograms. Regions of continuous nuclear membranes were traced in the 'non-core' region of the reforming NE. Three criteria were used to distinguish nascent NPCs from simple NE holes: (1) the gaps between the two membrane edges were less than 100 nm, (2) the two membrane edges tapered to a point suggesting there was fusion of the inner and outer NE, and (3) there were stretches of continuous membranes above and below the gap. To calculate the density of 'NPC' holes during NE reformation, we segmented and quantified four areas from a single control embryo (0.14 μm², 0.08 μm², 0.05 μm², and 0.04 μm²) and four areas from a single *ndc1Δ* embryo (0.22 μm², 0.19 μm², 0.18 μm², and 0.17 μm²). To calculate the density of 'NPC' holes during interphase, we segmented and quantified six regions (1.52 μm²,1.20 μm², 0.99 μm², 0.95 μm², 0.31 μm², and 0.19 μm²) from two different control nuclei and nine regions (1.03 μm², 1.02 μm², 0.87 μm², 0.65 μm², 0.58 μm², 0.55 μm², 0.52 μm², 0.41 μm², and 0.23 μm²) from *ndc1Δ* nuclei.

## Quantification and statistical analysis

### Image analysis

### Nuclear import analysis

To determine the fluorescence intensity of NLS-LacI::GFP inside the nucleus of one- and two-cell stage embryos, the chromatin was traced with either the freehand or circle tool in FIJI. Camera background was determined by drawing a 50 × 50 pixel box in vacant areas of the video. Average cytoplasmic values were determined by drawing a 20 × 20 pixel box inside the embryo. The nuclear to cytoplasmic ratio was determined by subtracting the average camera background from each value and then the nuclear value was divided by the cytoplasmic value. To account for differences in nuclear size, this ratio was then multiplied by the nuclear area. This process was repeated for each time point and condition.

### Line scan analysis of two-cell stage embryos

A three-pixel wide by 10 micron long line was drawn and centered on the nucleus to determine the fluorescence intensity. The line was then redrawn perpendicular to the first and a second measurement was taken. The values from these two lines were then averaged to give the fluorescence intensity of the NE. The average value of the first and last two points of each line was used to subtract background from the rest of the line. These values were then plotted against the relative position along the line.

### Line scan analysis of immunofluorescent images

A three-pixel wide by five-micron long line was drawn and centered on the NE. The line was then redrawn perpendicular to the first and a second measurement was taken. The values from these two lines were then averaged to give the fluorescence intensity of the NE. The same line scan was relocated to a clear area of the video to get the average intensity for camera background. Finally, to account for wide differences in fluorescence intensity, the data was normalized by dividing each value by the maximum fluorescence intensity. The final fluorescence intensities were then plotted against the relative position along the line. Additionally, the normalized value at the NE was divided by the first five values on the line to determine the NE:cytoplasmic ratio or by the last five values to determine the NE:nucleoplasmic ratio.

### Quantification of Nup160:GFP puncta and colocalization with Ndc1:mRuby

To determine the number of NPPGFP puncta/aggregates in the cytoplasm of one-cell stage embryos, maximum projections were generated using FIJI. The puncta were then tracked using the TrackMate plugin in FIJI (*Tinevez et al., 2017*). This process was repeated to determine the number of puncta in *ndc1* and *npp-7* RNAi-depleted embryos. To determine the colocalization between the Nup160:GFP puncta and Ndc1:mRuby puncta, three-pixel wide line scans were drawn in each channel and plots were overlayed.

### FRAP analysis

To evaluate the amount of recovery for Nup160:GFP and Ndc1:mNG, a rectangular ROI was used to measure the average fluorescence intensity of the NE before and after photobleaching. These values were normalized to the pre-bleach frame and graphed relative to bleaching. To calculate the mobile/immobile fraction as well as the T1/2, the data was fit to the exponential equation $f(t) = A \times (1 - \exp[-tau \times time])$. To obtain the best fit, the parameters were fitted to the data using non-linear least squares in R-studio.

### Pronuclear diameter measurement using MATLAB

Images captured on the Nikon spinning disc confocal microscope were converted into maximum projections with FIJI. Videos were saved as image sequences (.jpg) and read by the MATLAB script. Paternal pronuclei were segmented using a fluorescent threshold to transform the images into binary. Nuclear diameter was calculated using the MALTAB function 'regionprops' for each time frame. The surface area (formula) and volume ($4/3 \times pi \times r^3$) were calculated using the radius for the paternal pronucleus.

### Pronuclear volume calculation using MATLAB

Images captured on the diSPIM microscope were deconvolved using CytoShow to generate videos with isotropic resolution. The paternal pronucleus were then cropped in a 75 × 75 pixel box. For each time point and z-slice, the nuclear rim was tracked using the jFilament plugin in FIJI, and the coordinates of the ROI snake were saved. A folder containing the relevant data was read by MATLAB. For each nuclear slice, the geometric coordinates were used to calculate the polygonal area, which was then multiplied by 0.1625 µm (thickness of z-slice). The sum of these slices corresponded to the volume of the paternal pronucleus. For each time point, the widest slice of the nucleus was used to calculate the theoretical nuclear volume. This theoretical volume was then compared to the calculated volume to determine how spherical the pronuclei were. This analysis was performed for wild-type and *cnep-1Δ* embryos, and in both cases, the calculated values fell within 10% of the theoretical volume.

### Statistical analysis

All data points are reported in graphs and error bar types are noted in figure legends. Statistical analysis was performed on datasets with multiple samples and independent biological repeats. The type of test used, sample sizes, and p values are reported in figure legends or in text ($p<0.05$ defined as significant). Statistical tests were performed using GraphPad Prism.

## Acknowledgements

We thank Sarah Barger, Patrick Lusk, and Shoken Lee for helpful feedback on the manuscript and members of the Bahmanyar lab for helpful discussion. We thank Eric Hastie and Sarah Barger for help with cloning, the MBL Embryology course for insights into strain construction, and Jackson Gordon for help with FRAP analysis. We would like to thank Mark Moyle and Daniel Colón-Ramos (Yale School of Medicine) for help with the diSPIM imaging and 3D projections. We thank Peter Askjaer (Andalusian Center for Developmental Biology), Arshad Desai (Ludwig Institute for Cancer Research/UCSD), and Stephen Adam (Northwestern Feinberg School of Medicine) for generously sharing reagents and the Caenorhabditis Genetics Center and the Japanese knockout (KO) consortium for strains utilized in this study. This work was supported by an NSF CAREER Award to Shirin Bahmanyar (NSF CAREER 1846010), and NIH CMB Training Grant: T32GM007223-S1 to Michael Mauro.

## Additional information

### Funding

| Funder | Grant reference number | Author |
|---|---|---|
| National Science Foundation | 1846010 | Michael Sean Mauro<br>Gunta Celma<br>Shirin Bahmanyar |
| National Institutes of Health | T32GM007223-S1 | Michael Sean Mauro |
| National Institutes of Health | GM131004 | Shirin Bahmanyar |

The funders had no role in study design, data collection and interpretation, or the decision to submit the work for publication.

### Author contributions

Michael Sean Mauro, Conceptualization, Data curation, Formal analysis, Investigation, Methodology, Writing - original draft, Writing - review and editing; Gunta Celma, Formal analysis, Investigation; Vitaly Zimyanin, Magdalena M Magaj, Kimberley H Gibson, Data curation; Stefanie Redemann, Data curation, Investigation; Shirin Bahmanyar, Conceptualization, Funding acquisition, Methodology, Project administration, Resources, Supervision, Writing - original draft, Writing - review and editing

### Author ORCIDs

Michael Sean Mauro (iD) http://orcid.org/0000-0001-7013-6139

Kimberley H Gibson (iD) http://orcid.org/0000-0001-8881-1411
Stefanie Redemann (iD) http://orcid.org/0000-0003-2334-7309
Shirin Bahmanyar (iD) http://orcid.org/0000-0002-6583-5055

Decision letter and Author response
Decision letter https://doi.org/10.7554/eLife.75513.sa1
Author response https://doi.org/10.7554/eLife.75513.sa2

---

## Additional files

### Supplementary files
- Transparent reporting form
- Source code 1. Pronuclear tracker script.
- Source code 2. Volume calculator script.
- Source code 3. FRAP processing script.

### Data availability
All data generated or analyzed during this study are included in the manuscript and supporting files. Source data files have been included for all plots that contained multiple data points. EM tomograms have been submitted to Dryad https://doi.org/10.5061/dryad.kprr4xh6h.

The following dataset was generated:

| Author(s) | Year | Dataset title | Dataset URL | Database and Identifier |
|---|---|---|---|---|
| Mauro M, Celma G, Zimyanin V, Magaj M, Gibson K, Redemann S, Bahmanyar S | 2022 | Electron microscopy tomograms | https://dx.doi.org/10.5061/dryad.kprr4xh6h | Dryad Digital Repository, 10.5061/dryad.kprr4xh6h |

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
