## [Editor Report]

The authors elegant studies help understand how the process by which membranes needed for growth of nuclear envelop is coordinated with nuclear pore assembly during nuclear envelop assembly. This coordination is mediated by a transmembrane nucleoporin Ndc1.

---

## [Decision Letter]

[Editors' note: this paper was reviewed by Review Commons.]

---

## [Author Response]

Reviewer 1:Overall, this manuscript presents interesting and high quality data. However, there are several issues that the authors need to address prior to publication. Most importantly, some of the data presented appear contradictory, namely the decrease of different Nups at the NE upon Ndc1 RNAi and the unchanged mAb414 signal are not in line with the observed decrease in pore number. This should be discussed/studied in more detail. In addition, some important quantifications and controls are missing as specified below.Specific comments:1. Lines 111 and 121 and 179 and 200: better say "import" instead of "transport" (since export has not been formally addressed).

We have now changed the text throughout to indicate import rather than transport.

2. Line 130: "ndc1delta embryos contained small pronuclei (Figure 1B) there were similar in size to those RNAi-depleted for ndc1 (FigureS1C)". Quantifications are missing. Only the ndc1 RNAi pronuclei diameter is measured (Figure 1G), it would be good to have the size of the ndc1delta as well since this *C. elegans* strain is used throughout the paper and considered together with the Ndc1 RNAi embryo results to draw conclusions.

Analysis of the paternal pronuclear size of fixed one-cell stage embryos for control and ndc1 deleted embryos is now shown in new panel in Figure 1—figure supplement 1C. Note that this analysis had to be done on fixed one-cell stage embryos and so they represent a range of pronuclear diameters as compared to nuclear diameter measurements at a single time point in living embryos in Figure 1G. Overall, this analysis shows that pronuclei in both ndc1D and ndc1 RNAi-depletion are significantly smaller than pronuclei in control embryos.

3. Line 136: there are no quantifications of the difference in embryo sizes of Ndc1delta worms, and they do not always appear smaller throughout the figures of the paper (although the authors claim this in figure 1).

The reviewer is correct that the size of one-cell stage embryos is variable in the ndc1D worms. We now include measurements in fixed control and ndc1D one-cell stage embryos to provide a clearer representation of the range of embryo lengths we observe (see new lines #147-148).

4. Line 150: "pseudocleavage" à PC acronym is missing, which is used throughout the paper

We thank the reviewer for catching this omission and we have fixed the manuscript text. We have now indicated the acronym after the first use of the word “pseudocleavage” (new line #165) and use PC regression throughout to be consistent with the rest of the text.

5. The videos play back fast and slower speed would help.

Where appropriate, we have slowed the videos (see new Videos 1-4, 10, and 11), as suggested by the reviewer.

6. Line 165: "suggesting a role for Ndc1 is establishment of nuclear transport after mitosis (Anderson et al. 2009)." Typo.

We have omitted this sentence.

7. Figure 1F: Nice that the author show it so clearly but it is redundant and could go to supplementary material.

We have moved this Figure panel to new Figure 1-supplement 1H.

8. Line 172: "Ndc1en:mNG also localizes to the ER as well as punctate structures throughout the cytoplasm". These structures seem asymmetrically distributed. Are they coming from the mother cell, as it was shown for RanBP2 granules in *Drosophila* embryos (Hampoelz et al. 2019)? Do these Ndc1 granules colocalize with RanBP2 or other Nups? While this would add to the manuscript, in our opinion it is not strictly required for publication.

While we do agree that pursuing this line of research would likely yield interesting results, this would require the generation of strains and so we prefer to address these questions in a future study. We do provide some more direct interpretations of these puncta in relation to the role of Ndc1 in NPC assembly in the Discussion line #380-392.

9. Figure 2E: why does it decrease at the end?

We now clarify that the decrease signal in the nuclear NLS:GFP reporter at the end of this time course is because the embryo has entered mitosis, which triggers permeabilization of the nuclear envelope resulting in equilibration of the nuclear and cytoplasmic components. See new line #204-206.

10. Figure 2G: redundant information to Figure 2E, could be moved to supplementary material.

This panel has been moved to the supplement as the reviewer has suggested (see new Figure 2—figure supplement 1F). To avoid further redundancy, we also removed part of the previous Figure S2E, which showed the individual normalized traces for the entire time course, and only show the first 300s of the graph that represent the shift in onset of nuclear import.

11. The authors state that it was previously shown that Ndc1 delays NE reformation thus delaying the nuclear import at the end of mitosis (Anderson et al. 2009). But this is not the case here, since NE reformation seems to occur at the same time as in the control. This should be commented on and the differences should be discussed.

We have added these comments and the differences in the new Discussion pg. 17, lines 407-415. Additionally, we did not detect a difference in ER recruitment between wildtype and ndc1 RNAi-depleted embryos during NE formation (see Author response image 1).

**Author response image 1. sa2fig1:** 

12. Is the decrease in NPC density in ndc1delta embryos maintained beyond telophase? The authors could measure it in later time points by electron tomograms (or better by super resolution microscopy) and count the number of NPCs per square micrometer. A decrease is expected given that the export delays persist in later time points. Furthermore, analyzing NPC density will be important to support the tomography data that suggest a decreased number of NPCs but given the unchanged mAb414 signal.

We have now resolved the issue of the unchanged mAB414 signal in new Figure 3-figure supplment 2C (see also Response to Editor#1 and Reviewer1 #18 and #19). Importantly, we performed EM tomography on fully formed nuclei from multiple prepared samples of both control and ndc1D embryos at later time points (see new Figure 3—figure supplement 1A and B and new Results text lines #2227-231).

13. Figure 2D: color code explanation is missing. Some statistics are necessary, not only measurements of single cells

We have now added a clear color code to the figure legend to clarify the analysis (see new line #485). We also provide average value and standard deviation for when GFP:NLS are first observed in the nucleus in ndc1 RNAi-depleted embryos as well as the sample size to the text (see Results lines #201).

14. Lines 227-230: "The fact that they colocalize with Ndc1 and do not form in ndc1 RNAi embryos, which also have lower levels of p160:GFP at the nuclear rim, suggested that Ndc1 may play a role in stabilizing scaffold components both in these cytoplasmic structures and in the NE." ALs have been suggested to play a role in the postmitotic NPC assembly and to contribute to the rapid repopulation of NEs with NPCs (Ren et al. 2019). Since the cytoplasmic Nup60 granules do not form in the absence of ndc1, it could be that a defect in this pathway contributes to the decrease in NPCs in late mitosis. Please discuss or address

Thank you for this reference and suggestion. We now address this in the new Discussion on pg. 16, lines 380-392.

15. Line 234: the authors claim that "each component is approximately 50% less enriched at the nuclear rim in the absence of Ndc1 (Figure 3D)". Unfortunately, it is not explained how this 50% is calculated. This should be properly quantified

We have updated the text as to provide an explanation for how the 50% lower fluorescence signal at the nuclear rim was quantified in the absence of Ndc1. (See new line #245-250).

16. Figure 3D and 3G: ndc1delta embryo does not seem to be smaller than the ctrl in contradiction to the claim in the line 136 as discussed above.

Please see Reviewer1 #3.

17. Figure S3H: since the authors claim that Ndc1 regulates NPC assembly (and does not regulate Nup protein levels) it would be important to check that Nup levels are not decreased, especially for the member of the Y complex ELYS (to make sure that postmitotic assembly is not being indirectly affected by Ndc1 through ELYS). Checking an importin would also be a nice control, to show that the effect on import is indeed due to decrease in NPC number

We would like to thank the reviewer for these suggestions. We now show that NPC number is lower in both the reforming nuclei (Figure 3A and 3B) and at later time points in fully formed nuclei (see new Figure 3—figure supplement 1A and B and new Results text line #227-231) indicating defects in NPC assembly result from loss of Ndc1. We now also show a reduction in mAB414-epitope containing nucleoporins signal in ndc1D embryos but overall levels are unchanged (see new Figure 3—figure supplement 2C and Reviewer1 #18 and #19) corroborating our results that there is a defect in NPC biogenesis without Ndc1. Based on the Reviewer’s suggestion, we added new data in new Figure 3—figure supplement 2F showing that the total protein levels of the main importin expressed in early embryos IMA-3 are unchanged between control and ndc1D worms further suggesting that the delay and reduced nuclear import is not due to reduced transport machinery. We were unable to reliably detect a band specifically corresponding to Elys after several attempts of using an endogenous antibody generously provided by Arshad Desai’s lab (Hattersley et al. 2016) as well as a GFP antibody in a strain expressing ELYS:GFP.

18. The quality of the mAb414 WB on FigureS4D is not good. We expect to see 4 bands (corresponding to C.e. Nup358, Nup214, Nup153 and Nup62). Please repeat WB with tris acetate gel and wet transfer to see high molecular size proteins and check levels.

We purchased a new mAB414 antibody from a different company that clarified these discrepancies. Using this new antibody, we address the issues raised by this Reviewer in several ways.

1. As suggested by the Reviewer, we used a 3 – 8% Tris-Acetate gel with wet transfer to run lysate prepared from *C. elegans* (see new Figure 3—figure supplement 2D) and mammalian cells (see Author response image 2). In lysates prepared from mammalian cells, we detected 4 prominent bands recognized by mAB414 antibodies at the predicted molecular weights for Nup358 (Ce npp-4), Nup214 (Ce npp-14), Nup153 (Ce npp-7) and Nup62 (Ce npp-11), which are 95 kDa, 148 kDa, 124 KDa, and 77 kDa, respectively, for the *C. elegans* proteins (see Author response image 2). However, mAB414 antibodies recognized 3 protein species between 75 kDa and ~120 kDa in lysates prepared from *C. elegans* (see new Figure 3—figure supplement 2D and Author response image 2). Thus, mAB414 detects distinct protein species in *C. elegans*, as has also been shown in Galy et al. 2003.

2. Based on Galy et al. 2003, one of the *C. elegans* proteins recognized by mAB414 antibodies is Ce Nup98. We also show that the disappearance of the Nup98 protein species following RNAi-depletion and immunoblotting of worm lysates (see new Figure 3—figure supplement 2D) indicating that the mAB414 antibody is specific for at least Nup98 and likely other nucleoporins in *C. elegans*.

19. Line 241-243: "Immunostaining of control and ndc1 mutant worms with mAB414, a monoclonal antibody that recognizes FG-nucleoporins, did not show a consistent difference in fluorescence signal at the nuclear rim in early stage ndc1 mutant embryos (Figure S4C)". How do the authors explain this? If there are fewer NPCs, mAb414 signal should also be affected. It is critical that the authors address this point. Furthermore, figure S4C shows embryos in different (more advanced?) stages than the ones imaged throughout the rest of the paper. Would this potentially explain the mAb414 levels? Are the decreased in earlier time points? NPC density quantifications at later time points would be useful.

We have fixed this technical issue by using a new fixation protocol and the new mAB414 antibody (see Response to Editor #1 and Response to Reviewer1#18). We now show that the fluorescence signal of mAB414-epitope containing nucleoporins at the NE is reduced in ndc1D worms (see new Figure 3—figure supplement 2C). This result together with our results using EM tomography, antibodies against other nucleoporins, and fluorescent fusion proteins (see Figure 3, Figure 3—figure supplement 1 and 2) corroborates our overall conclusion that loss of ndc1 results in reduced NPCs at the nuclear rim.

20. Figure 3E: SUN1 has been shown to be associated with NPCs upon assembly during the interphasic pathway. Maybe worth to check if Ndc1 helps to stabilize NPC dynamics through interaction with the LINC complex? Maybe SUN1 recruitment to NE could be analyzed in presence and absence of Ndc1.

This is a very interesting idea, however, we believe this experiment is out of the scope of this manuscript because of the lack of characterization of the interphase NPC assembly pathway in early *C. elegans* embryos.

21. Line 275: "Live imaging of nup53tm2886 one-cell embryos revealed that sperm pronuclei expand and establish import (Figure 4C), similar to nup53 RNAi-depleted nuclei (Figure 1E-1G) but to a lesser extent than control and ndc1 RNAi-depleted embryos (Figure 1E-1G)". Not precisely correct, it should rather be: Live imaging of nup53tm2886 one-cell embryos revealed that sperm pronuclei establish import (Figure 4C), and pronuclear expansion is similar to nup53 RNAi-depleted nuclei and ndc1delta (Figure 1E-1G) but less than control and ndc1 RNAi-depleted embryos (Figure 1E-1G).

The text has been changed to clarify as the reviewer requested on new line #324.

22. Figure 4E and G: Why is the graph inverted graph compared to figure 1G. This is confusing.

We changed the graphs (see new Figure 5E and G).

23. Line 325: "The cytoplasmic puncta we observed with Ndc1 may represent annulate lamellae, specialized membranes in the ER that are prominent in embryos and serve to couple NPC assembly with NE formation (Hampoelz et al. 2016), or the holes in the ER containing pre-assembled nuclear pore scaffold components that serve to template post-mitotic NPC assembly (Chou et al. 2021)." Or cytoplasmic Nup condensates maternally given prior to AL formation (Hampoelz et al. 2019). Please include a discussion on the asymmetry of the granules and refer to the Hampoelz paper

We have now reference to the Hampoelz et al. 2019 paper in this context. See line #s 109-114 and in the Discussion line #374-385.

24. Line 337: Please add data and comment on the relevance if this NPC density defect is transitory or persists during the cell cycle and/or development.

We have re-written the Discussion in a way that incorporates our new data showing the lower NPC numbers persist throughout the cell cycle (see new Discussion pp. 16-17). See also Response to Reviewer1 #12 and Response to Reviewer 2 #2

Reviewer 2 Major Comments:1. The observation that recruitment of the NUP107-NUP160 complex is strongly impaired in NDC1-depleted embryos is unexpected based on the Mansfeld study mentioned above ("The NE association of Nup107, a component of the Nup107-160 subcomplex essential for NPC biogenesis was slightly reduced"; Mansfeld et al., 2006). One explanation could be that the depletion of NDC1 is more efficient in *C. elegans* (known for its potent RNAi machinery). Alternatively, although many metazoans have three integral membrane nuclear pore proteins (NUP210, POM121, NDC1; as stated in line 69), POM121 seems to be absent in *C. elegans*. The Mansfeld study reported that cells co-depleted for POM121 and NDC1 had more severe phenotypes than the single depletions. Perhaps *C. elegans* NDC1 carry out the functions of both vertebrate proteins (POM121 and NDC1) and thus the depletion produces a stronger effect. Although this is speculative, I suggest the authors to discuss this – or at least mention in line 69 that POM121 is not detected in *C. elegans*.

We thank the reviewer for bringing this point to our attention. We agree that it is important to provide more information on POM121 and the fact that it is not detected in *C. elegans* may account for the differences we observe in these systems. We have now added new text to address this point in the new Introduction on p. 2-3, pg. 5 lines #118-123 and in the new Discussion on pp. 17 line #407-415.

2. The use of electron tomograms to evaluate NE formation is exciting. This is a powerful technique that to my knowledge has not been used before for this purpose in *C. elegans* embryos. However, more details on the analysis and more observations are needed. For instance, how big are the areas that were quantified? Are the areas (2) in control; 3 areas in ndc1(0) from a single embryo? Even if 2 control and 3 ndc1(0) embryos were used, these are very low numbers and carry a potential risk that the embryos are not perfectly staged. NE and NPC reformation is an extremely fast process in *C. elegans* embryos and small variations in fixation time could introduce a bias in the number of NPCs/holes in the tomograms.

We appreciate the reviewer seeing the novelty of this approach to evaluate NPC assembly upon nuclear reformation. Below we address your comments/suggestions:

i. How big are the areas that were quantified? Are the areas (2) in control; 3 areas in ndc1(0) from a single embryo? We quantified 4 areas from a single control embryo (0.14 mm^2^, 0.08 mm^2^, 0.05 mm^2^, and 0.04 mm^2^) and 4 areas from a single ndc1D embryo (0.22 mm^2^, 0.19 mm^2^, 0.18 mm^2^, and 0.17 mm^2^). We now clarify this in the main text and both in the Figure legend pg. 26 and in the Materials and methods pg. 49.

ii. We agree with the Reviewer’s concern regarding staging of embryos. We now provide data collected from tomographs of nuclei in two cell stage embryos increasing our sample size and the number of nuclei that we analyzed (see new text in Results line #227-231, new Figure 3—figure supplement 1A and B, and new Materials Methods pp. 49). These data reveal a consistent decrease in NPC density in fully formed nuclei corroborating our immunofluorescence and live imaging data (see Figure 3D, Figure 3—figure supplement 1E and F, Figure 3—figure supplement 2A and C) as well as the FRAP analysis of Nup160:GFP (see Figure 3F), reduced import rates and smaller nuclear size measurements (see Figure 1F and new Figure 4 and Figure 4—figure supplement 1). The addition of these data together with the addition of new Figure 4 provides evidence for a role for Ndc1 in NPC assembly during nuclear formation and expansion.

3. I'm concerned regarding the conclusion on Ndc1 mobility based on the FRAP assays presented. The nucleus in Figure 3G seems to grow in size during the course of the FRAP experiment. This growth presumably involves incorporation of ER membranes that might bring in new NPP-22 (not necessarily into NPCs; hence the difference with Nup106 in Figure 3E). The FRAP should be performed on fully grown nuclei although this will leave less time before the next cell division. Potentially could be performed in non-dividing cells. Because the authors state several times – including in the title – that Ndc1 is a dynamic protein, this is a critical point.

We thank this Reviewer for bringing up this critical point. The Reviewer is correct that the incorporation of new Ndc1::mNG-associated membrane could affect the dynamics of Ndc1 recovery that we report. The dynamics of Ndc1 no longer serves as a major point with the new addition of new Figure 4 and Figure 4—figure supplement 1 that focuses on membrane addition and NPC assembly via Ndc1. Nevertheless, we performed FRAP on oocyte nuclei and found that Ndc1:mNG is immobile in these NEs (see new Figure 3E). Thus, we removed the conclusion that Ndc1 is dynamic at the NE from the manuscript title and throughout the manuscript. We also moved the original data to the supplement (see new Figure 3—figure supplement 2G), and changed the Results text to read:

“It has been proposed that transmembrane nucleoporins serve as an anchor to immobilize the NPC in the NE. In nuclei that were expanding, we found that Ndc1^en^:mNG was highly mobile in the NE (Figure 3—figure supplement 2G and Video 12; average mobile fraction ± SD = 0.59 ±- 0.09, n = 7 embryos). This suggests that Ndc1^en^:mNG may dynamically associate with nascent NPCs. However, growth of nuclei could result in recovery of Ndc1^en^:mNG through feeding of a new pool of membrane-associated Ndc1 and so we tested the turnover of Ndc1^en^:mNG in the fully expanded nuclei of oocytes. Ndc1^en^:mNG turnover was slow in oocyte NEs indicating that Ndc1^en^:mNG was immobile in mature NPCs (Figure 3E; average mobile fraction ± SD = 0.22 ±- 0.2, n = 9 embryos). Thus, NDC1 may serve an anchoring role for mature NPCs.” (Results lines # 266-275)

We also address this in the Discussion lines #401-406, which now reads:

“Our data showing that Ndc1 may be dynamic in growing NEs supports recent data using metabolic labeling in budding yeast showed that Ndc1, unlike other transmembrane nucleoporins, is readily exchanged in the NPC (Hakhverdyan et al., 2021; Onischenko et al., 2020).. However, we also found Ndc1 is highly stable in fully formed nuclei suggesting it serves as an anchor to mature NPCs. Future mechanistic work is required to determine the mechanism by which Ndc1 drives NPC assembly.”

Minor comments:4. Gene and protein names are not used consistently. For instance, in line 233, *C. elegans* protein names are added in superscript (Nup133NPP-15, etc.) but not when Nup160 is introduced the first time. In lines 156-157, *C. elegans* gene names are instead given in brackets. In Similarly, in the Key Resources table, the "official" WormBase name npp-22 is used, whereas ndc-1 is used several times in the text.

The text has been changed throughout the manuscript and in the key resource table.

5. Lines 44-45: "The NE compartmentalizes the genome…" Probably more correct to say that the NE compartmentalizes the cell.

We changed this sentence.

6. Line 264: "the Nup96/98 complex". Which complex(es) are the authors referring to? NUP96 is part of the Nup107-NUP160 complex but NUP98 isn't.

The reviewer is correct and this was our mistake. The text has been changed accordingly throughout.

Reviewer 3:Major commentsThe key conclusions are convincing and the experimental data of high quality. From my side no additional experiments are suggested. The experiments are sufficiently replicated and statistical analysis is adequate. The text and figures are clear and accurate and the story is easy to follow.Minor comments:1. Line 72: "there is relatively little is known" – please correct.

We thank the Reviewer for catching this textual typo. The text has been fixed.

2. Line 72: Given the amount of work done on NDC1, POM121, and NDC1 in metazoan the statement that relatively little is known about their organization and function: NDC1 and POM121 are involved in NPC assembly, GP210 in NPC breakdown and muscle differentiation, POM121 in prostate cancer progression to name just a few. It would be fair to mention at least some of the work done.

We agree with the reviewer and have significantly added to the introduction and Discussion to address this point. Please new Introduction on p. 2-3, pg. 5 lines #118-123 and in the new Discussion on pp. 17 line #409-415. Also see Response to Reviewer2 #1.

3. Line 84: Otsuka et al. do not show that Y-complex and IRC components associate with fenestrations in the ER during mitosis.

We thank the reviewer for catching this mistake. We have fixed this (see new lines # 107-108).

4. Line 85: I don't understand the flow of arguments. Why do the authors refer here to AL in *Drosophila*? How does this connect to the next sentence?

We now added a new topic sentence to this paragraph to help the logical flow (see new line #103).

5. Line 102f: "Prior work using…": the Sentence could be split to be grammatically correct.

We fixed this.

6. Line 112: "In line with these findings…": I find this result surprising (as the authors) but do not understand why this "in line with these findings".

We fixed this.

7. Line 120: The sentence is an overstatement: NDC1 could regulate establishment of nuclear transport also by other means, e.g. assembly of the IRC, which would be crucial for central channel FG-repeat nucleoporin assembly.

We agree with the Reviewer that Ndc1 could be acting through directly assembly of the IRC. However, although we do not understand the mechanistic connection, our FRAP analysis shows higher mobile fraction of GFP:Nup160 in the NE results from loss of Ndc1. Thus, we argue that our data shows a role for Ndc1 in stable incorporation of the outer ring scaffold. We have changed the sentence to read:

“Our data suggest that Ndc1 controls NPC density by promoting stable incorporation of outer ring scaffold components.”

8. Line 227 and line 327: Would be good to mention here that the Hampoelz et al. refer to *Drosophila* embryos.

We removed this sentence and reference *Drosophila* embryos in the context of Hampoelz paper.

9. Line 254: wrong citation , do you refer to Rabut et al. (PMID: 15502822)?

We thank the reviewer for catching the mistaken citation. We have now altered the text to correct the citation. See new line #280.

10. Line 264: I don't know a Nup96/98 complex. Do you mean Nup93?

This was a typo. We have changed this sentence completely and define the inner ring Nup93 complex in the first paragraph of the introduction.

11. Line 306: "We go on show" – please correct.

We removed this sentence.

12. Line 327: In my eyes, the cited work shows rather that ALs contribute to NE expansion, not formation.

We agree with the reviewer that the cited work most accurately reflects a role of AL’s in nuclear expansion not formation. We have changed the text throughout to reflect this change.

13. Figure Legend 4 E and G: error bars are SD?

We thank the reviewer for catching this omission. The reviewer is correct, the error bars are in fact standard deviation (SD). The text has been fixed in the figure legend to clarify that the error bars are SD.

Referee Cross-commentingReviewer 3: Reviewer 2 raises a very valid point about the FRAP analysis: As the mobility of NDC1 is a novel and in my eyes most interesting aspect of the manuscript it would be good to confirm these in non-dividing cells.

We found that Ndc1::mNG is indeed immobile in fully grown oocyte nuclei that are non-dividing (see new Figure 3E). We cannot exclude the possibility that the higher mobility pool of Ndc1::mNG in NEs of embryos is faster than at mature NPCs and so we removed the original data from the main figure and discuss the caveats of the experiment, as pointed out by Reviewer 2 (see also Response to Reviewer2 #3). We also have removed this point from the title, abstract and discussion.

As agreed to by the editors at *eLife*, we include a new Figure 4 and new Figure 4—figure supplement 1 that addresses the role of Ndc1-mediated NPC assembly and regulation of membrane biogenesis in nuclear reformation and size control. We believe that the addition of these data and the reframing of the manuscript with these new data increases the significance and impact of the manuscript’s findings.

Reviewer 3: Regarding the point raised by Reviewer 4, the connection between NPC numbers and nuclear size: In tissue culture cells, RNAi against nucleoporins often reduces nuclear sizes but I am not sure whether this has been systematically analyzed.

We now show that upregulated membrane biogenesis restores the small nuclear size resulting from loss of Ndc1, but does not restore NPCs or import rates (see new Figure 4 Figure 4—figure supplement 1, and Video 13). Furthermore, excess membranes do not restore small nuclear size resulting from loss of Nup53 or Nup153. These data suggest a potentially distinct role for Ndc1 in coupling NPC assembly to membrane incorporation and provide insight into the connection between NPC numbers and nuclear size.

Reviewer 4: Generally, the *C. elegans* NPC is not well-studied in the field, however, the surprising non-essential phenotype of the *C. elegans* NDC1 partial knock-out strain (tm1845) has been previously established by the Goerlich laboratory (Stavru et al. JCB 2006). Using primarily fluorescence microscopy experiments and various genetically modified *C. elegans* strains that build on these previous results, the authors confirm the previous non-lethal phenotype of the tm1845 strain and by electron microscopy analysis confirm the previous conclusion that the partial NDC1 knock-out yields nuclear envelopes with vastly reduced NPC numbers. Using CRISPR/CAS9, the authors generated a complete knock-out of the NDC1 coding region, demonstrating a more severe but still viable phenotype. I am unaware whether the reported small nuclei phenotype has been observed previously in other organisms, but I find the proposed NPC number to nuclear size connection interesting, but unfortunately this aspect of the manuscript is not well developed.

We have added new data (see new Figure 4, Figure 4—figure supplement 1, Video 13) in the revised manuscript demonstrating that the smaller nuclear size of ndc1 RNAi-depleted embryos can be rescued to near wild type levels by increasing membrane biogenesis. Importantly, increased membrane biogenesis restores nuclear size, but not reduced NPC levels or nuclear import rates resulting from loss of ndc1. Furthermore, increased membrane biogenesis does not rescue nuclear size in nup53 or nup153 RNAi-depleted embryos (new Figure 4J) suggesting that there is a distinct relationship between membrane incorporation and NPC assembly via Ndc1. Together, our data suggests that Ndc1 functions in parallel to Nup53 and membrane biogenesis to determine NPC density and nuclear size. This work goes well beyond the Stavru et al. paper on Ndc1 in *C. elegans* and provides an explanation for why Ndc1 is non-essential in worms.

Reviewer 4: The NPC schematics in Figure 4A does not represent the current state of knowledge. The authors should modify the location of the various analyzed proteins as reported in the cited reviews (Lin and Hoelz, Annu Rev Biochem 2019, Hampoelz, Andres-Pons, Kastritis, Beck, Annu Rev Biophysics, 2019). It would be useful for a reader outside of the field to show the location of all analyzed nups in this schematic and perhaps would be best included in Figure 1 (Y complex nucleoporins, Nup53, Nup153, and Ndc1).

We thank the reviewer for pointing out this error. We have altered the schematic to address this issue (see new Figure 5A).